# Is Noise Conditioning Necessary for Denoising Generative Models?

Qiao Sun [* 1]   Zhicheng Jiang [* 1]   Hanhong Zhao [* 1]   Kaiming He [1]

## Abstract

It is widely believed that noise conditioning is indispensable for denoising diffusion models to work successfully. This work challenges this belief. Motivated by research on blind image denoising, we investigate a variety of denoising-based generative models in the absence of noise conditioning. To our surprise, most models exhibit graceful degradation, and in some cases, they even perform better without noise conditioning. We provide a theoretical analysis of the error caused by removing noise conditioning and demonstrate that our analysis aligns with empirical observations. We further introduce a noise-*unconditional* model that achieves a competitive FID of 2.23 on CIFAR-10, significantly narrowing the gap to leading noise-conditional models. We hope our findings will inspire the community to revisit the foundations and formulations of denoising generative models.

## 1. Introduction

At the core of denoising diffusion models (Sohl-Dickstein et al., 2015) lies the idea of corrupting clean data with *various* levels of noise and learning to reverse this process. The remarkable success of these models has been partially underpinned by the concept of "*noise conditioning*" (Sohl-Dickstein et al., 2015; Song & Ermon, 2019; Ho et al., 2020): a single neural network is trained to perform denoising across all noise levels, with the noise level provided as a conditioning input. The concept of noise conditioning has been predominantly incorporated in diffusion models and is widely regarded as a critical component.

In this work, we examine the necessity of noise conditioning in denoising-based generative models. Our intuition is that, in natural data such as images, the noise level can be reliably

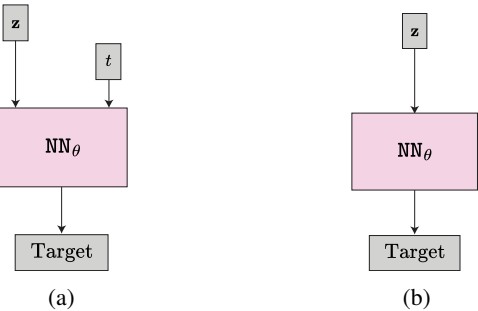

Figure 1: **(a)** A denoising generative model takes a noisy data $\mathbf{z}$ and a noise level indexed by $t$ (such as $\sigma_t$) as the inputs to the neural network $\text{NN}_{\boldsymbol{\theta}}$. **(b)** This work investigates the scenario of removing noise conditioning in the network.

estimated from corrupted data, making *blind* denoising (*i.e.*, without knowing the noise level) a feasible task. Notably, noise-level estimation and blind image denoising have been active research topics for decades (Stahl et al., 2000; Salmeri et al., 2001; Rabie, 2005), with neural networks offering effective solutions (Chen et al., 2018; Guo et al., 2019; Zhang et al., 2023). This raises an intriguing question: can related research on image denoising be generalized to denoising-based generative models?

Motivated by this, in this work, we systematically compare a variety of denoising-based generative models — *with and without* noise conditioning. Contrary to common belief, we find that many denoising generative models perform robustly even in the absence of noise conditioning. In this scenario, most methods exhibit only a modest degradation in generation performance. More surprisingly, we find that some relevant methods—particularly flow-based ones (Lipman et al., 2023; Liu et al., 2023), which originated from different perspectives—can even produce *improved* generation results *without* noise conditioning. Among all the popular methods we studied, only one variant fails disastrously. Overall, our empirical results reveal that noise conditioning may *not* be necessary for denoising generative models to function properly.

We present a theoretical analysis of the behavior of these models in the absence of noise conditioning. Specifically, we investigate the inherent uncertainty in the noise level distribution, the error caused by denoising without noise condi-

---

*Equal contribution. Listing order is random. [1]MIT. Correspondence to: Qiao Sun <sqa24@mit.edu>, Zhicheng Jiang <jzc_2007@mit.edu>, Hanhong Zhao <zhh24@mit.edu>, Kaiming He <kaiming@mit.edu>.

*Proceedings of the $42^{nd}$ International Conference on Machine Learning*, Vancouver, Canada. PMLR 267, 2025. Copyright 2025 by the author(s).

tioning, and the accumulated error in the iterated sampler. Put together, we formulate an error bound that can be computed without involving any training, depending solely on the noise schedules and the dataset. Experiments show that this error bound correlates well with the noise-unconditional behaviors of the models we studied—particularly in cases where the model fails catastrophically, its error bound is orders of magnitudes higher.

Because noise-unconditional models have been rarely considered, it is worthwhile to design models specifically for this underexplored scenario. To this end, we present a simple alternative derived from the EDM model (Karras et al., 2022). Without noise conditioning, our variant can achieve a strong performance, reaching an FID of 2.23 on the CIFAR-10 dataset. This result significantly narrows the gap between a noise-unconditional system and its noise-conditional counterpart (*e.g.*, 1.97 FID of EDM).

Looking ahead, we hope that removing noise conditioning will pave the way for new advancements in denoising-based generative modeling. For example, only in the absence of noise conditioning can a score-based model learn a unique score function and enable the classical, physics-grounded Langevin dynamics.[1] Overall, we hope that our findings will motivate the community to re-examine the fundamental principles of related methods and explore new directions in the area of denoising generative models.

## 2. Related Work

**Noise Conditioning.**  The seminal work of diffusion models (Sohl-Dickstein et al., 2015) proposes iteratively perturbing clean data and learning a model to reverse this process. In this pioneering work, the authors introduced a "*time dependent readout function*", which is an early form of noise conditioning.

The modern implementation of noise conditioning is popularized by the introduction of *Noise Conditional* Score Networks (NCSN) (Song & Ermon, 2019). NCSN is originally developed for score matching. This architecture is adopted and improved in Denoising Diffusion Probabilistic Models (DDPM) (Ho et al., 2020), which explicitly formulate generation as an iterative denoising problem. The practice of noise conditioning has been inherited in iDDPM (Nichol & Dhariwal, 2021), ADM (Dhariwal & Nichol, 2021), and nearly all subsequent derivatives.

DDIM (Song et al., 2021a) and EDM (Karras et al., 2022) reformulate the reverse diffusion process into an ODE solver, enabling deterministic sampling from a single initial noise. Flow Matching (FM) models (Lipman et al., 2023; Liu et al.,

2023; Albergo et al., 2023) reformulate and generalize the framework by learning flow fields that map one distribution to another. In all these methods, noise conditioning (also called time conditioning) is the *de facto* choice.

Beyond diffusion models, Consistency Models (Song et al., 2023) have emerged as a new family of generative models for non-iterative generation. It has been found (Song & Dhariwal, 2024) that noise conditioning and its implementation details are critical for the success of consistency models, highlighting the central role of noise conditioning.

**Blind Image Denoising.**  In the field of image processing, blind image denoising has been studied for decades. It refers to the problem of denoising an image without any prior knowledge about the level, type, or other characteristics of the noise. Relevant studies include noise level estimation from noisy images (Stahl et al., 2000; Shin et al., 2005; Liu et al., 2013; Chen et al., 2015), as well as directly learning to perform blind denoising from data (Liu et al., 2007; Chen et al., 2018; Batson & Royer, 2019; Zhang et al., 2023). Modern neural networks, including the U-Net (Ronneberger et al., 2015) commonly used in diffusion models, have been shown highly effective for these tasks.

Our research is closely related to classical work on blind denoising. However, the iterative nature of the generative process, where errors can accumulate, introduces new challenges. In addressing these challenges, our work opens up new research opportunities that extend classical approaches.

## 3. Formulation

In this section, we present a reformulation that can summarize the training and sampling processes of various denoising generative models. The core motivation of our reformulation is to *isolate* the neural network $\text{NN}_{\boldsymbol{\theta}}$, allowing us to focus on its behavior with respect to noise conditioning.

### 3.1. Denoising Generative Models

**Training Objective.**  During training, a data point $\mathbf{x}$ is sampled from the data distribution $p(\mathbf{x})$, and a noise $\boldsymbol{\epsilon}$ is sampled from a noise distribution $p(\boldsymbol{\epsilon})$, such as a normal distribution $\mathcal{N}(\mathbf{0}, \boldsymbol{I})$. A noisy image $\mathbf{z}$ is given by:

$$\mathbf{z} = a(t)\mathbf{x} + b(t)\boldsymbol{\epsilon}. \qquad (1)$$

Here, $a(t)$ and $b(t)$ are schedule functions that are method-dependent. The time step $t$, which can be a continuous or discrete scalar, is sampled from $p(t)$. Without loss of generality, we refer to $b(t)$, or simply $t$, as the *noise level*.

In general, a denoising generative model involves minimizing a loss function that can be written as:

$$\mathcal{L}(\boldsymbol{\theta}) = \mathbb{E}_{\mathbf{x},\boldsymbol{\epsilon},t}\Big[w(t)\big\|\text{NN}_{\boldsymbol{\theta}}(\mathbf{z}|t) - r(\mathbf{x},\boldsymbol{\epsilon},t)\big\|^2\Big]. \quad (2)$$

---

[1]Otherwise, it relies on the *annealed* Langevin dynamics (Song & Ermon, 2019)) that does not correspond to a unique underlying probability distribution independent of noise levels.

Table 1: Schedules used by different models in our reformulation. Notations and details are in Appendix D.

|  | iDDPM, DDIM | EDM | FM |
|---|---|---|---|
| $a(t)$ | $\sqrt{\bar{\alpha}(t)}$ | $\frac{1}{\sqrt{t^2+\sigma_d^2}}$ | $1-t$ |
| $b(t)$ | $\sqrt{1-\bar{\alpha}(t)}$ | $\frac{t}{\sqrt{t^2+\sigma_d^2}}$ | $t$ |
| $c(t)$ | $0$ | $\frac{t}{\sigma_d\sqrt{t^2+\sigma_d^2}}$ | $-1$ |
| $d(t)$ | $1$ | $-\frac{\sigma_d}{\sqrt{t^2+\sigma_d^2}}$ | $1$ |

Here, $\text{NN}_{\theta}$ is a neural network (*e.g.*, U-Net) to be learned, $r(\mathbf{x}, \boldsymbol{\epsilon}, t)$ is a *regression target*, and $w(t)$ is a weight. The regression target $r$ can be written as:

$$r(\mathbf{x}, \boldsymbol{\epsilon}, t) = c(t)\mathbf{x} + d(t)\boldsymbol{\epsilon}, \tag{3}$$

where $c(t)$ and $d(t)$ are also method-specific schedule functions. Common choices of $r$ include $\boldsymbol{\epsilon}$-prediction (Ho et al., 2020), $\mathbf{x}$-prediction (Salimans & Ho, 2022), or $\mathbf{v}$-prediction (Salimans & Ho, 2022; Lipman et al., 2023).

The specifics of the schedule functions of several existing methods are in Table 1. It is worth noting that, in our reformulation, we concern the regression target $r$ with respect to the neural network $\text{NN}_{\theta}$'s *direct* output.[2]

**Sampling.** Given trained $\text{NN}_{\theta}$, the sampler performs iterative denoising. Specifically, with an initial noise $\mathbf{x}_0 \sim \mathcal{N}(0, b(t_{\max})^2 \boldsymbol{I})$, the sampler iteratively computes:

$$\mathbf{x}_{i+1} := \kappa_i \mathbf{x}_i + \eta_i \text{NN}_{\theta}(\mathbf{x}_i | t_i) + \zeta_i \tilde{\boldsymbol{\epsilon}}_i. \tag{4}$$

Here, a discrete set of time steps $\{t_i\}$ is pre-specified and indexed by $0 \leqslant i < N$. The schedules, $\kappa_i$, $\eta_i$, and $\zeta_i$, can be computed from the training-time noise schedules in Table 1 (see their specific forms in Appendix D). In Eq. (4), $\tilde{\boldsymbol{\epsilon}}_i \sim \mathcal{N}(\mathbf{0}, \boldsymbol{I})$ is a sampling-time noise that only takes effect in SDE-based solvers; there is no noise added in ODE-based solvers, *i.e.*, $\zeta_i = 0$.

Eq. (4) is a general formulation that can encapsulate many first-order samplers, such as (annealed) Langevin sampling and Euler-based ODE solver. Higher-order samplers (*e.g.*, Heun) can be formulated similarly with extra schedules. In this paper, our theoretical analysis is based on Eq. (4), and higher-order cases are evaluated empirically.

### 3.2. Noise Conditional Networks

In existing methods, the neural network $\text{NN}_{\theta}(\mathbf{z}|t)$ is conditioned on the noise level specified by $t$. See Fig. 1

---

[2]For methods like EDM where the network output is wrapped with a precondition, we rewrite the schedules to expose the term of $\text{NN}_{\theta}$ (see Appendix D.3). This network $\text{NN}_{\theta}$ is called the "*raw network*" in EDM (see Eq. (8) in (Karras et al., 2022)).

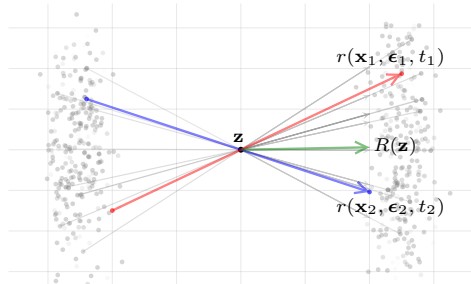

Figure 2: **Illustration of the effective target $R(\mathbf{z})$.** A given $z$ corresponds to multiple triplets $(\mathbf{x}, \boldsymbol{\epsilon}, t)$. Here, we take Flow Matching (Lipman et al., 2023) as an example. On the left are the samples of $\boldsymbol{\epsilon}$, and on the right are samples of $\mathbf{x}$. For a noisy sample $\mathbf{z} = (1 - t)\mathbf{x} + t\boldsymbol{\epsilon}$, it can be produced by different triplets. Each triplet gives a different regression target $r$. The effective target $R(\mathbf{z})$ is the expectation of all possible $r$.

(left). This is commonly implemented as $t$-embedding, such as Fourier (Tancik et al., 2020) or positional embedding (Vaswani, 2017). This $t$-embedding provides time-level information as an additional input to the network. Our study concerns the influence of this noise conditioning, that is, we consider $\text{NN}_{\theta}(\mathbf{z})$ vs. $\text{NN}_{\theta}(\mathbf{z}|t)$. See Fig. 1 (right). Note that $\text{NN}_{\theta}(\mathbf{z})$ or $\text{NN}_{\theta}(\mathbf{z}|t)$ involves all learnable parameters in the model, while the schedules ($a(t)$, $b(t)$, *etc.*) are pre-designed and not learned.

## 4. Analysis of Noise-Unconditional Models

Based on the above formulation, we present a theoretical analysis of the influence of removing noise conditioning. Our analysis involves both the training objectives and the sampling process. We first analyze the effective target of regression at the training stage and its error in a single denoising step (Sections 4.1 to 4.3), and then give an upper bound on the accumulated error in the iterative sampler (Section 4.4). Overall, our analysis provides an error bound that is to be examined by experiments.

### 4.1. Effective Targets

While the loss function is often written in a form like Eq. (2), the underlying *unique* regression target for $\text{NN}_{\theta}(\mathbf{z}|t)$ is **not** $r(\mathbf{x}, \boldsymbol{\epsilon}, t)$. The function $\text{NN}_{\theta}(\mathbf{z}|t)$, which is w.r.t. $\mathbf{z}$ and $t$, is regressed onto *multiple* $r$ values corresponding to different possible triplets $(\mathbf{x}, \boldsymbol{\epsilon}, t)$ that produce the same $\mathbf{z}$ (see Fig. 2). Intuitively, the unique effective target, denoted as $R(\mathbf{z}|t)$ to emphasize its dependence on $\mathbf{z}$ and $t$, is the expectation of $r$ over all possible triplets.

Formally, optimizing the loss in Eq. (2) is equivalent to optimizing the following loss, where each term inside the

expectation $\mathbb{E}[\cdot]$ has a unique effective target:

$$\mathcal{L}(\boldsymbol{\theta}) = \mathbb{E}_{\mathbf{z}\sim p(\mathbf{z}),t\sim p(t|\mathbf{z})}\Big[\big\|\mathrm{NN}_{\boldsymbol{\theta}}(\mathbf{z}|t) - R(\mathbf{z}|t)\big\|^2\Big]. \quad (5)$$

Here, $p(\mathbf{z})$ is the marginalized distribution of $\mathbf{z}:=a(t)\mathbf{x} + b(t)\boldsymbol{\epsilon}$ in Eq. (1), under the joint distribution $p(\mathbf{x},\boldsymbol{\epsilon},t) := p(\mathbf{x})p(\boldsymbol{\epsilon})p(t)$.[3] It is easy to show that:

$$R(\mathbf{z}|t) = \mathbb{E}_{(\mathbf{x},\boldsymbol{\epsilon})\sim p(\mathbf{x},\boldsymbol{\epsilon}|\mathbf{z},t)}\big[r(\mathbf{x},\boldsymbol{\epsilon},t)\big], \quad (6)$$

that is, the expectation over all $(\mathbf{x},\boldsymbol{\epsilon})$ subject to the conditional distribution. One can show (Appendix C.1) that minimizing Eq. (5) is equivalent to minimizing Eq. (2), and similar analysis has also been done in previous work(Lehtinen et al., 2018).

**Effective Targets without Noise Conditioning.** Similarly, if the network $\mathrm{NN}_{\boldsymbol{\theta}}(\mathbf{z})$ does not accept $t$ as the condition, its unique effective target $R(\mathbf{z})$ should depend on $z$ only. In this case, the loss is:

$$\mathcal{L}(\boldsymbol{\theta}) = \mathbb{E}_{\mathbf{z}\sim p(\mathbf{z})}\Big[\big\|\mathrm{NN}_{\boldsymbol{\theta}}(\mathbf{z}) - R(\mathbf{z})\big\|^2\Big], \quad (7)$$

where the unique effective target is:

$$R(\mathbf{z}) = \mathbb{E}_{t\sim p(t|\mathbf{z})}\big[R(\mathbf{z}|t)\big]. \quad (8)$$

Eq. (8) suggests that if the conditional distribution $p(t|\mathbf{z})$ is close to a Dirac delta function, the *effective target* would be the same with and without conditioning on $t$. If so, assuming the network is capable enough to fit the target, the noise-unconditional variant would produce the same output as the conditional one.

### 4.2. Concentration of Posterior $p(t|\mathbf{z})$

Next, we investigate how similar $p(t|\mathbf{z})$ is to a Dirac delta function. For *high-dimensional* data such as images, it has been long realized that the noise level can be reliably estimated (Stahl et al., 2000; Salmeri et al., 2001; Shin et al., 2005), implying a concentrated $p(t|\mathbf{z})$. We note that the concentration of $p(t|\mathbf{z})$ depends on data dimensionality:

**Statement 1 (Concentration of $p(t|\mathbf{z})$).** *Consider a single datapoint $\mathbf{x} \in [-1,1]^d$, $\boldsymbol{\epsilon}\sim\mathcal{N}(\mathbf{0},\mathbf{I})$, $t\sim\mathcal{U}[0,1]$, and $\mathbf{z} = (1-t)\mathbf{x} + t\boldsymbol{\epsilon}$ (the Flow Matching case). Given a noisy image $\mathbf{z} = (1-t_*)\mathbf{x} + t_*\boldsymbol{\epsilon}$ produced by a given $t_*$, the variance of $t$ under the conditional distribution $p(t|\mathbf{z})$, is:*

$$\mathrm{Var}_{t\sim p(t|\mathbf{z})}[t] \approx \frac{t_*^2}{2d}, \quad (9)$$

*when the data dimension $d$ satisfies $\frac{1}{d} \ll t_*$ and $\frac{1}{d} \ll 1-t_*$. (Derivation in Appendix C.2)*

---
[3]For simplicity, we consider $w(t)=1$, which happens to be the case for all methods in Table 1 when we expose $\mathrm{NN}_{\boldsymbol{\theta}}$ explicitly.

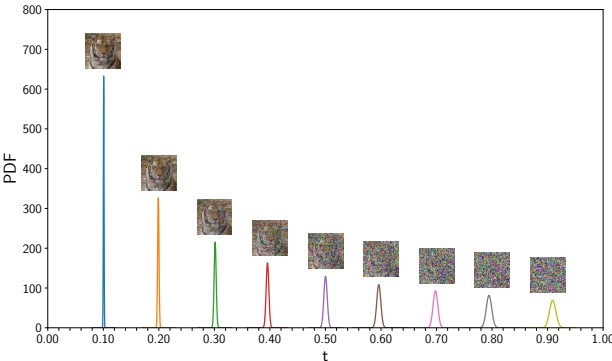

Figure 3: **The Posterior distribution $p(t|\mathbf{z})$ is concentrated.** We picked $\mathbf{z} = (1-t_*)\mathbf{x} + t_*\boldsymbol{\epsilon}$ with $t_*$ from 0.1 to 0.9 for illustration. This plot is empirically simulated from 15,000 images in the AFHQ-v2 dataset with a size $64\times64$ (see Appendix A.2).

Intuitively, this statement suggests that *high-dimensional* data induces a *sharply peaked* $p(t \mid \mathbf{z})$. In Appendix C.2, we derive a rigorous upper bound on this variance and extend the analysis to the multi–data–point setting. To corroborate these theoretical findings, we empirically run a simulation on a real dataset and plot $p(t|\mathbf{z})$ (see Fig. 3). The empirical distribution of $p(t|\mathbf{z})$ is well concentrated. Moreover, a smaller $t^*$ leads to a more concentrated $p(t|\mathbf{z})$, as also indicated by Eq. (9).

### 4.3. Error of Effective Regression Targets

With $p(t|\mathbf{z})$, we investigate the error between the effective regression targets $R(\mathbf{z})$ and $R(\mathbf{z}|t)$. Formally, we consider:

$$E(\mathbf{z}) := \mathbb{E}_{t\sim p(t|\mathbf{z})}\Big[\big\|R(\mathbf{z}|t) - R(\mathbf{z})\big\|^2\Big]. \quad (10)$$

We show that this error $E(\mathbf{z})$ is substantially smaller than the norm of $R(\mathbf{z})$:

**Statement 2 (Error of effective regression targets).** *Consider the scenario in Statement 1 and the Flow Matching case. The error defined in Eq. (10) satisfies:*

$$E(\mathbf{z}) \approx \frac{1}{2}(1 + \sigma_{\mathrm{d}}^2) \quad (11)$$

*when the data dimension $d$ satisfies $\frac{1}{d} \ll t_*$ and $\frac{1}{d} \ll 1-t_*$. Here, $\sigma_{\mathrm{d}}$ denotes the per-pixel standard deviation of the dataset. (Derivation in Appendix C.3)*

Intuitively, Statement 2 suggests that for sufficiently high-dimension $d$, the error $E(\mathbf{z})$ is substantially smaller ($\approx 1$) than the L2 norm of the target $R(\mathbf{z})$ ($\approx d$). In our real-data verification, we find that $E(\mathbf{z})$ is at the order of $1/10^3$ of $R(\mathbf{z})$ (see Appendix A.2). In this case, regressing to $R(\mathbf{z}|t)$ can be reliably approximated by regressing to $R(\mathbf{z})$.

## 4.4. Accumulated Error in Sampling

Thus far, we have been concerned with the error of a single regression step. In a denoising generative model, the sampler at inference time is iterative. We investigate the accumulated error in the iterative sampler.

To facilitate our analysis, we assume the network $\text{NN}_{\boldsymbol{\theta}}$ is sufficiently capable of fitting the effective regression target $R(\mathbf{z}|t)$ or $R(\mathbf{z})$. Under this assumption, we replace $\text{NN}_{\boldsymbol{\theta}}$ in Eq. (4) with $R$. This leads to the following statement:

**Statement 3 (Bound of accumulated error).** *Consider a sampling process, Eq. (4), of $N$ steps, starting from the same initial noise $\mathbf{x}_0 = \mathbf{x}'_0$. With noise conditioning, the sampler computes:*

$$\mathbf{x}_{i+1} = \kappa_i \mathbf{x}_i + \eta_i R(\mathbf{x}_i|t_i) + \zeta_i \tilde{\boldsymbol{\epsilon}}_i,$$

*and without noise conditioning, it computes:*

$$\mathbf{x}'_{i+1} = \kappa_i \mathbf{x}'_i + \eta_i R(\mathbf{x}'_i) + \zeta_i \tilde{\boldsymbol{\epsilon}}_i.$$

*Assuming $\|R(\mathbf{x}'_i|t_i) - R(\mathbf{x}_i|t_i)\| \, / \, \|\mathbf{x}'_i - \mathbf{x}_i\| \leqslant L_i$ and $\|R(\mathbf{x}'_i) - R(\mathbf{x}'_i|t_i)\| \leqslant \delta_i$, it can be shown that the error between the sampler outputs $\mathbf{x}_N$ and $\mathbf{x}'_N$ is bounded:*

$$\|\mathbf{x}_N - \mathbf{x}'_N\| \leqslant A_0 B_0 + A_1 B_1 + \ldots + A_{N-1} B_{N-1}, \quad (12)$$

*where:*

$$A_i = \prod_{j=i+1}^{N-1} (\kappa_i + |\eta_i| L_i) \quad and \quad B_i = |\eta_i| \delta_i.$$

*depend on the schedules and the dataset. (Derivation in Appendix C.4)*

Here, the assumption on $\delta_i$ can be approximately satisfied as per Statement 2. The assumption on $L_i$ models the function $R(\cdot|t)$ as Lipschitz-continuous. Although it is unrealistic for this assumption to hold exactly in real data, we empirically find that an appropriate choice of $L_i$ can ensure the Lipchitz condition holds with high probability (Appendix A.3).

Statement 3 suggests that the *schedules* $\kappa_i$ and $\eta_i$ are influential to the estimation of the error bound. With different schedules across methods, their behavior in the absence of noise conditioning can be dramatically different.

**Discussions.** Remarkably, the bound estimation can be computed *without* training the neural networks: it can be evaluated solely based on the schedules and the dataset.

Furthermore, our analysis of the "error" bound implies that the noise-conditional variant is more accurate, with the noise-*unconditional* variant striving to approximate it. In fact, there is no reason to assume that the former should be a more accurate generative model. Nonetheless, in experiments, we find that the noise-*unconditional* case can outperform its noise-conditional counterpart in some cases.

## 5. A Noise Unconditional Diffusion Model

In addition to investigating existing models, we also design a diffusion model specifically tailored for noise *unconditioning*. Our motivation is to find schedule functions that are more robust in the absence of noise conditioning, while still maintaining competitive performance. To this end, we build upon the highly effective EDM framework (Karras et al., 2022) and modify its schedules.

A core component of EDM is a "preconditioned" denoiser:

$$c_{\text{skip}}(t)\hat{\mathbf{z}} + c_{\text{out}}(t)\text{NN}_{\boldsymbol{\theta}}\big(c_{\text{in}}(t)\hat{\mathbf{z}} \mid t\big)$$

Here, $\hat{\mathbf{z}} := \mathbf{x} + t\boldsymbol{\epsilon}$ is the noisy input before the normalization performed by $c_{\text{in}}(t)$,[4] which we simply set as $c_{\text{in}}(t) = \frac{1}{\sqrt{1+t^2}}$. The main modification we adopt for the noise *unconditioning* scenario is to set:

$$c_{\text{out}}(t) = 1.$$

As a reference, EDM set $c_{\text{out}}(t) = \frac{\sigma_{\text{d}} t}{\sqrt{\sigma_{\text{d}}^2 + t^2}}$ where $\sigma_{\text{d}}$ is the data std. As $c_{\text{out}}(t)$ is the coefficient applied to $\text{NN}_{\boldsymbol{\theta}}$, we expect setting it to a constant will free the network from modeling a $t$-dependent scale. In experiments (Section 6.2), this simple design exhibits a lower error bound (Statement 3) than EDM. We name this model as **uEDM**, which is short for *(noise-)unconditional EDM*. For completeness, the resulting schedules of uEDM are provided in Appendix D.5.

## 6. Experiments

**Experimental Settings.** We empirically evaluate the impact of noise conditioning across a variety of models:

- **Diffusion**: iDDPM (Nichol & Dhariwal, 2021), DDIM (Song et al., 2021a), ADM (Dhariwal & Nichol, 2021), EDM (Karras et al., 2022), and uEDM (Sec. 5)

- **Flow-based Models**: we adopt the implementation of Rectified Flow (1-RF) (Liu et al., 2023), which is a form of Flow Matching (Lipman et al., 2023) (FM).

- **Consistency Models**: iCT (Song & Dhariwal, 2024) and ECM (Geng et al., 2025).

Our main experiments are on class-unconditional generation on CIFAR-10 (Krizhevsky et al., 2009), with extra results on ImageNet 32×32 (Deng et al., 2009), and FFHQ 64×64 (Karras et al., 2019). We evaluate Fréchet Inception Distance (FID) (Heusel et al., 2017) and report Number of Function Evaluations (NFE). For a fair comparison, all methods are based on our re-implementation as faithful as possible (see Appendix B.3): with and without noise conditioning are run in the same implementation for each method.

---

[4]To make notations consistent with our reformulation in Eq. (2), we denote $\mathbf{z} = c_{\text{in}}(t)\hat{\mathbf{z}}$. See details in Appendix D.3.

Table 2: **Changes of FID scores in the absence of noise conditioning**, for different methods on CIFAR-10. Here 'w/o $t$' means without noise conditioning. A color of yellow denotes a non-disastrous (and often decent) degradation; green denotes improvement; red denotes failure.

| model | sampler | NFE | FID w/ $t$ | $\rightarrow$ | w/o $t$ |
|---|---|---|---|---|---|
| iDDPM | SDE | 500 | 3.13 | $\rightarrow$ | 5.51 |
| iDDPM ($\mathbf{x}$-pred) | SDE | 500 | 5.64 | $\rightarrow$ | 6.33 |
| | ODE | 100 | 3.99 | $\rightarrow$ | 40.90 |
| DDIM | SDE | 100 | 8.07 | $\rightarrow$ | 10.85 |
| | SDE | 1000 | 3.18 | $\rightarrow$ | 5.41 |
| ADM | SDE | 250 | 2.70 | $\rightarrow$ | 5.27 |
| EDM | Heun | 35 | **1.99** | $\rightarrow$ | 3.36 |
| | Euler | 50 | 2.98 | $\rightarrow$ | 4.55 |
| | Euler | 100 | 3.01 | $\rightarrow$ | 2.61 |
| FM (1-RF) | Heun | 99 | 2.87 | $\rightarrow$ | 2.63 |
| | RK45 | $\sim$127 | 2.53 | $\rightarrow$ | 2.63 |
| iCT | - | 2 | 2.59 | $\rightarrow$ | 3.57 |
| ECM | - | 2 | 2.57 | $\rightarrow$ | 3.27 |
| uEDM (Sec. 5) | Heun | 35 | 2.04 | $\rightarrow$ | **2.23** |

## 6.1. Main Observations

Table 2 summarizes the FID changes in different generative models, with and without noise conditioning, denoted as "w/ $t$" and "w/o $t$". Fig. 5 shows some qualitative results. We draw the following observations:

**(i)** Contrary to common belief, noise conditioning is *not* an enabling factor for most denoising-based models to function properly. Most variants can work gracefully, exhibiting small but decent degradation ( yellow ).

**(ii)** More surprisingly, some flow-based variants can achieve *improved* FID ( green ) after removing noise conditioning. In general, flow-based methods investigated in this paper are insensitive to whether we use noise conditioning or not. We hypothesize that this is partially because FM's regression target is independent on $t$ (see Table 1: $c = -1$, $d = 1$)

**(iii)** The uEDM variant (Sec. 5) achieves a competitive FID of 2.23 without noise conditioning, narrowing the gap to the strong baseline of the noise-conditional methods (here, 1.99 of EDM, or 1.97 reported in Karras et al. (2022)).

**(iv)** Consistency Models (here, iCT and ECM), which are related to diffusion models but present a substantially different objective function, can also perform gracefully. While iCT was found highly sensitive to the subtleties of $t$-conditioning (see Song & Dhariwal (2024)), we find that removing it does not lead to disastrous failure.

**(v)** Among all variants we investigate, only "DDIM w/ ODE sampler" results in a catastrophic failure ( red ), with FID significantly worsened to 40.90. Fig. 5 (a) demonstrates its

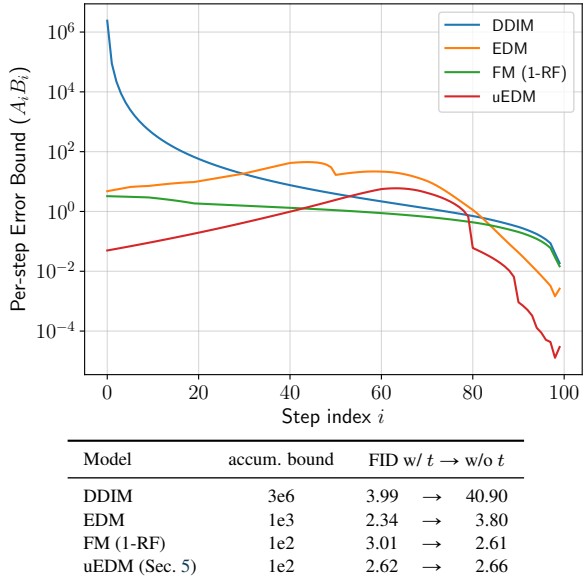

| Model | accum. bound | FID w/ $t \rightarrow$ w/o $t$ | | |
|---|---|---|---|---|
| DDIM | 3e6 | 3.99 | $\rightarrow$ | 40.90 |
| EDM | 1e3 | 2.34 | $\rightarrow$ | 3.80 |
| FM (1-RF) | 1e2 | 3.01 | $\rightarrow$ | 2.61 |
| uEDM (Sec. 5) | 1e2 | 2.62 | $\rightarrow$ | 2.66 |

Figure 4: **Error bound and the influence of noise conditioning.** ODE with $N = 100$ steps is applied for each variant. The plot shows the per-step error bound $A_i B_i$ in Eq. (12), and the table shows the accumulated error bound. The y-axis is log-scale.

qualitative behavior: the model *is still able* to make sense of shapes and structures; it is "overshoot" or "undershoot", producing over-saturated or noisy results.

**Summary.** Our experimental findings highlight that *noise conditioning, though often helpful for improving quality, is not essential for the fundamental functionality of denoising generative models*.

## 6.2. Analysis

**Error Bound.** In Fig. 4, we empirically evaluate the error bound in Statement 3 for different methods under a 100-step ODE sampler. The computation of the bound depends only on the schedules for each methods, as well as the dataset (detailed in Appendix A.3).

Fig. 4 shows a strong correlation between the theoretical bound and the empirical behavior. Specifically, DDIM's catastrophic failure can be explained by its error bound that is orders of magnitudes higher. On the other hand, EDM, FM, and uEDM all have small error bounds throughout. This is consistent with their graceful behavior in the lack of noise conditioning.

These findings suggest that the error bound derived in our analysis serves as a reliable predictor of a model's robustness to the removal of noise conditioning. Importantly, the bound can be computed solely based on the model's formulation and dataset statistics, *without* training the neural

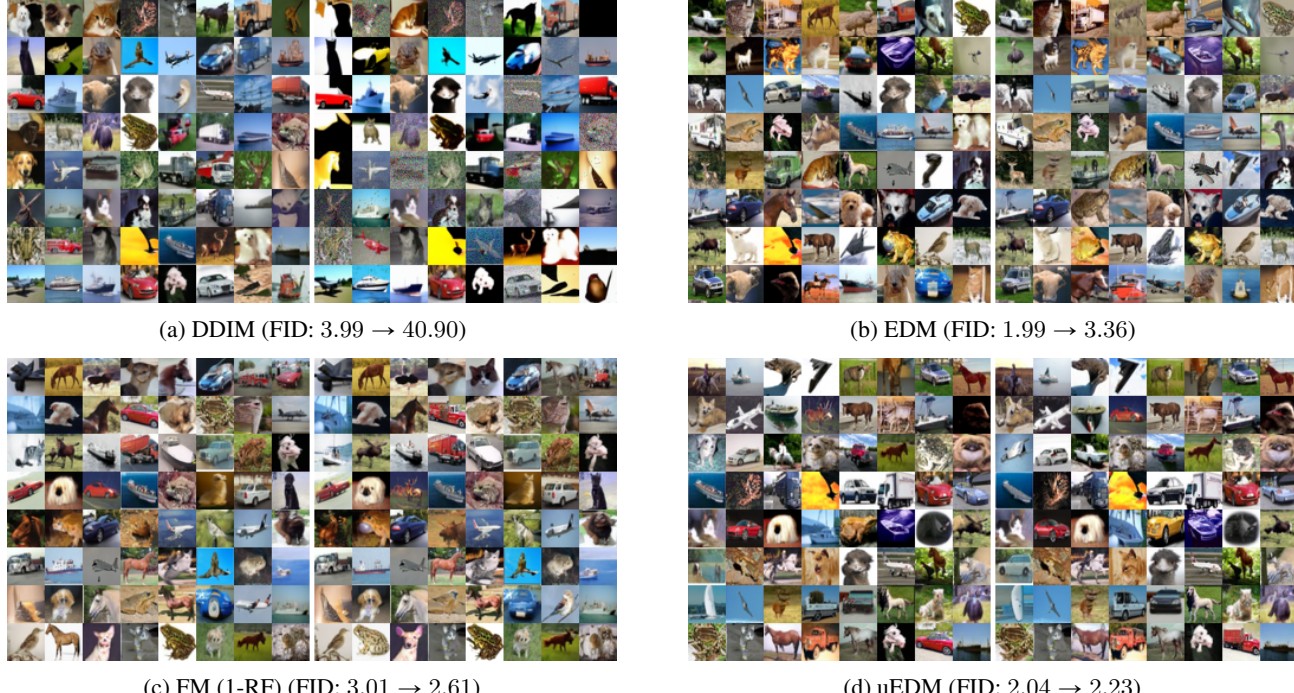

(a) DDIM (FID: 3.99 → 40.90)  (b) EDM (FID: 1.99 → 3.36)

(c) FM (1-RF) (FID: 3.01 → 2.61)  (d) uEDM (FID: 2.04 → 2.23)

Figure 5: **Samples of noise-conditional vs. noise-unconditional models.** Samples are generated by (a) DDIM, (b) EDM, (c) FM (1-RF), and (d) uEDM, on the CIFAR-10 class-unconditional case. For each subfigure, the left panel is the noise-conditional case, and the right panel is the noise-unconditional counterpart, with the same random seeds. The change of FID is from "w/ $t$" to "w/o $t$". See also Table 2 for more quantitative results.

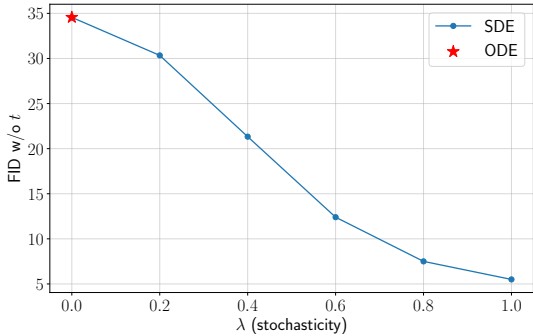

Figure 6: **Influence of Stochasticity on DDIM**, in the lack of noise conditioning. The level of stochasticity is specified by $\lambda$, with $\lambda = 0$ denoting the ODE case. Here, the number of sampling steps is fixed as 500.

network. Consequently, it can provide a valuable tool for estimating whether a given denoising generative model can function effectively without noise conditioning, prior to model training.

**Level of Stochasticity.** In Table 2, DDIM only fails with the deterministic ODE sampler (the default sampler in (Song et al., 2021a)); it still performs decently with the SDE sampler (*i.e.*, the DDPM sampler). With this observation, we

further investigate the level of stochasticity in Fig. 6.

Specifically, with the flexibility of DDIM (Song et al., 2021a), one can introduce a parameter $\lambda$ that interpolates between the ODE and SDE samplers by adjusting $\eta_i$ and $\zeta_i$ in Eq. (4) (see Eq. (56) in Appendix D.2). As shown in Fig. 6, increasing $\lambda$ (more stochasticity) consistently improves FID scores. When $\lambda = 1$, DDIM behaves similarly to iDDPM.

We hypothesize that this phenomenon can be explained by error propagation dynamics. Our theoretical bound in Statement 3 assumes worst-case error accumulation, but in practice, stochastic sampling enables error cancellation. The ODE sampler's consistent noise patterns lead to correlated errors, while the SDE sampler's independent noise injections at each step promote error cancellation. This error cancellation mechanism can improve performance with increasing stochasticity, as further evidenced by iDDPM and ADM's results (Table 2) produced by SDE.

**Alternative Noise-conditioning Scenarios.** Thus far, we have focused on removing noise conditioning from existing models. This is analogous to blind image denoising in the field of image process. Following the research topic on noise level estimation, we can also let the network explicitly or implicitly predict the noise level. Specifically, we consider the following four cases (Fig. 7):

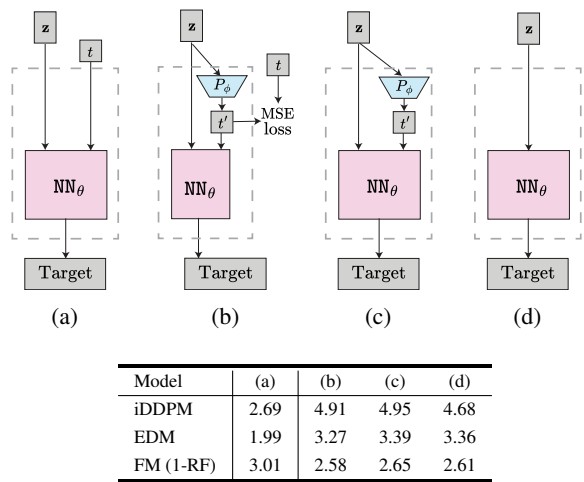

| Model | (a) | (b) | (c) | (d) |
|---|---|---|---|---|
| iDDPM | 2.69 | 4.91 | 4.95 | 4.68 |
| EDM | 1.99 | 3.27 | 3.39 | 3.36 |
| FM (1-RF) | 3.01 | 2.58 | 2.65 | 2.61 |

Figure 7: **Alternative Noise-conditional Scenarios. (a)** Noise-conditional baseline. **(b)** Noise-conditional, but on $t'$ predicted by a noise level predictor $P$. **(c)** Similar to (b), but the noise level predictor is not supervised and is trained jointly. **(d)** Noise un-conditional baseline. For iDDPM, EDM, and FM, all of (b), (c), and (d) perform similarly.

**(a)** The standard noise-conditioning baseline, which is what we have been comparing with. See Fig. 7(a).

**(b)** A noise-conditioning variant, in which the noise level is predicted by another network. In this variant, the noise predictor $P$ is a small network pre-trained to regress $t$. This predictor is then frozen when training $\text{NN}_{\theta}$, and $\text{NN}_{\theta}$ is conditioned on the predicted $t'$, rather than the true $t$. See Fig. 7(b).

**(c)** An "unsupervised" noise-conditioning variant. This architecture is exactly the same as the variant (b), except that the noise predictor $P$ is trained from scratch without any ground-truth $t$. If we consider $P$ and $\text{NN}_{\theta}$ jointly as a larger network, this also represents a design for noise-unconditional modeling. See Fig. 7(c).

**(d)** The standard noise-unconditional baseline, which is what we have been investigating. See Fig. 7(d).

Fig. 7 compares all four variants. Notably, consistent behavior is observed for all models (iDDPM, EDM, and FM) studied here: the results of (b), (c), and (d) are similar. This suggests that (b), (c), and (d) could be *subject to the same type of error*, that is, the uncertainty of $t$ estimation. Note that even in the case of (b) where the noise predictor is pre-trained with the true $t$ given, its prediction cannot be perfect due to the small yet inevitable uncertainty in $p(t|\mathbf{z})$ (see Section 4.2). As a result, the supervised pre-trained noise predictor (b) does not behave much different with the unsupervised counterpart (c).

Table 3: Changes of FID scores in the absence of noise conditioning, on class-unconditional ImageNet 32×32 and FFHQ 64×64, and class-conditional CIFAR-10.

| Model | Sampler | NFE | FID w/ $t$ → w/o $t$ |
|---|---|---|---|
| **ImageNet 32×32** | | | |
| FM (1-RF) | Euler | 100 | 5.15 → 4.85 |
| **FFHQ 64×64** | | | |
| EDM | Heun | 79 | 2.64 → 3.59 |
| **CIFAR-10 Class-conditional** | | | |
| EDM | Heun | 35 | 1.76 → 3.11 |
| FM (1-RF) | Euler | 100 | 2.72 → 2.55 |

### 6.3. Extra Datasets and Tasks.

Thus far, our experiments have been on the CIFAR-10 class-unconditional task. To show the generalizability of our findings, we further evaluate class-unconditional generation on ImageNet 32×32, FFHQ 64×64, and class-conditional generation on CIFAR-10. See Table 3.

The behavior is in general similar to that in our previous experiments. Specifically, removing noise conditioning can also be effective for other datasets or the class-conditional generation task. FM can exhibit *improvement* in the absence of noise conditioning; EDM has a decent degradation, but experience no catastrophic failure.

### 6.4. Classifier-Free Guidance

We further examine the impact of omitting noise conditioning when using classifier-free guidance (CFG) (Dhariwal & Nichol, 2021), a standard technique for significantly improving sample quality in class-conditional diffusion models.

Our experiments with CFG are conducted on the ImageNet 256×256 dataset on SiT (Ma et al., 2024), which is a flow-matching variant of DiT (Peebles & Xie, 2023). For comparison, we train SiT-B/2 under the original paper's configuration for both noise-conditional and unconditional model. At inference, we employ an Euler sampler with 250 steps and vary the CFG scale. See more details in Appendix B.

The results in Fig. 8 indicate that removing the noise conditioning incurs almost no degradation at different guidance scales, corroborating our analysis.

## 7. Discussion and Conclusion

We hope that rethinking the role of noise conditioning will open up new opportunities. Modern diffusion models are closely related to Score Matching (Hyvärinen & Dayan, 2005; Song & Ermon, 2019; Song et al., 2021b), which provides an effective solution to Energy-Based Models (EBM) (Hopfield, 1982; Ackley et al., 1985; LeCun et al., 2006; Song & Kingma, 2021). The key idea of EBM is to represent a probability distribution $p(x)$ by $p(x) = e^{-E(x)}/Z$, where

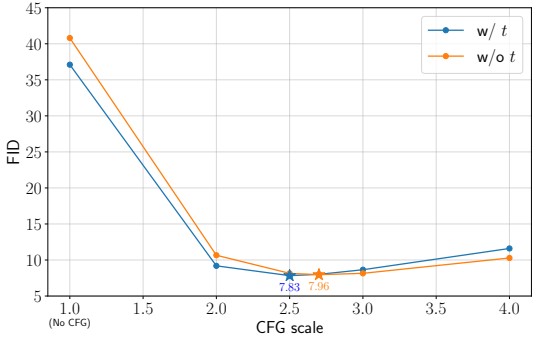

Figure 8: **Classifier-free guidance results for SiT-B/2 on ImageNet** $256 \times 256$**.** We use an Euler sampler with 250 steps. The best guidance scale in each setting is marked with a star. All other experimental details follow the original paper. Removing noise conditioning leads to almost no degradation regardless of guidance scale.

$E(x)$ is the energy function. With the score function of $p(x)$ (that is, $\nabla_x E(x)$), one can sample from the underlying $p(x)$ by Langevin dynamics. This classical formulation models the data distribution $p(x)$ by a *single* energy function $E(x)$ that is solely dependent on $x$. Therefore, a classical EBM is inherently $t$-unconditional. However, with the presence of $t$-conditioning, the sampler becomes *annealed* Langevin dynamics (Song & Ermon, 2019), which implies *a sequence* of energy functions $\{E(x, t)\}_t$ indexed by $t$, with one sampling step performed on each energy. Our study suggests that it is possible to pursue a *single* energy function $E(x)$, aligning with the goal of classical EBM.

Our study also reveals that certain families of models, *e.g.*, Flow Matching (Lipman et al., 2023; Liu et al., 2023; Albergo et al., 2023), can be more robust to the removal of $t$-conditioning. Although these models are closely related to diffusion, they can be formulated from a substantially different perspective—estimating a flow field between two distributions. While these models can inherit the $t$-conditioning design of diffusion models, their native formulation does not require the flow field to be dependent on $t$. Our study suggests that there exists a *single* flow field for these methods to work effectively.

In summary, noise conditioning has been predominant in modern denoising-based generative models and related approaches. We encourage the community to explore new models that are not constrained by this design.

## Acknowledgements

We greatly thank Google TPU Research Cloud (TRC) for granting us access to TPUs. Q. Sun, Z. Jiang, and H. Zhao are supported by the MIT Undergraduate Research Opportunities Program (UROP). We thank Cheng Jiang, Yiyang Lu, Mingyang Deng, Xingjian Bai, and Xianbang Wang for their comments and discussions.

## Impact Statement

This paper presents work whose goal is to advance the field of Machine Learning. There are many potential societal consequences of our work, none which we feel must be specifically highlighted here.

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

# Contents

# Appendices

## A. Details of Numerical Experiments

In this section, we provide additional details on all our real dataset numerical experiments. By first computing the value of some relevant quantities (*e.g.* the underlying time distribution $p(t|\mathbf{z})$, effective target $R(\mathbf{z}|t)$, $R(\mathbf{z})$), we are able to evaluate $E(\mathbf{z})$, which is average error introduced by removing noise conditioning. See Appendix A.1.

As we introduce single data point assumption in our theoretical framework, we verify the accuracy of this assumption by comparing the empirical values of $p(t|\mathbf{z})$ and $E(\mathbf{z})$ with the theoretical values derived from our estimations. See Appendix A.2.

Finally, in Appendix A.3, we show how we derive the numbers in Fig. 4 by detailing on our estimation of bound values $A_i$ and $B_i$ in Statement 3.

### A.1. Computation of Relavent Quantities

We consider the data distribution $p_{\text{data}}$ constituted solely from the data points in the dataset: $p_{\text{data}}(\mathbf{x}) = \frac{1}{N}\sum_{i=1}^{N}\delta(\mathbf{x}-\mathbf{x}_i)$, where $\mathbf{x}_i \in \mathbb{R}^d$ are the images in the dataset, and $\delta(\cdot)$ is the delta distribution. We denote $N$ as the number of data points in the dataset, and $d$ as the dimension of the image.

**Calculation of $p(t|\mathbf{z})$ (Section 4.2).**   First, we calculate $p(\mathbf{z}|t)$ by marginalizing over all the data points:

$$p(\mathbf{z}|t) = \int p(\mathbf{z}|t,\mathbf{x})p(\mathbf{x})\,\mathrm{d}\mathbf{x} = \frac{1}{N}\sum_{i=1}^{N} p(\mathbf{z}|t,\mathbf{x}_i).$$

The random variable $\mathbf{z}$ is given by Eq. (1), which implies

$$p(\mathbf{z}|t,\mathbf{x}) = \mathcal{N}\left(\mathbf{z}; a(t)\mathbf{x}, b(t)^2\boldsymbol{I}_d\right), \tag{13}$$

where $\mathcal{N}(\mathbf{z};\boldsymbol{\mu},\boldsymbol{\Sigma})$ denotes the probability density function of the Gaussian distribution with mean $\boldsymbol{\mu}$ and covariance $\boldsymbol{\Sigma}$. This leads to

$$p(\mathbf{z}|t) = \frac{1}{N}\sum_{i=1}^{N}\mathcal{N}\left(\mathbf{z}; a(t)\mathbf{x}_i, b(t)^2\boldsymbol{I}_d\right), \tag{14}$$

and we can finally obtain:

$$p(t|\mathbf{z}) = \frac{p(t)}{p(\mathbf{z})}p(\mathbf{z}|t) = \frac{p(t)\sum\limits_{i=1}^{N}\mathcal{N}\left(\mathbf{z}; a(t)\mathbf{x}_i, b(t)^2\mathbf{I}_d\right)}{\int_0^1 p(t)\sum\limits_{i=1}^{N}\mathcal{N}\left(\mathbf{z}; a(t)\mathbf{x}_i, b(t)^2\mathbf{I}_d\right)\mathrm{d}t}. \tag{15}$$

Note that there is an integral to evaluate in Eq. (15). In practice, the calculation is performed in a two-step manner for a fixed $\mathbf{z}$. In the first step, we use a uniform grid of $100$ $t$ values in $[0,1]$ (*i.e.* $t = 0.00, 0.01, \ldots, 0.99$). We calculate the value of $p(t)p(\mathbf{z}|t)$ for each $t$ value.

Typically, we observe that within an interval $[l, r]$ (where $0 \leqslant l < r \leqslant 1$), the value of $p(t)p(\mathbf{z}|t)$ is significantly larger than for other $t \in [0,1]$ [5]. We then approximate the integral as:

$$\int_0^1 p(t)\sum\limits_{i=1}^{N}\mathcal{N}\left(\mathbf{z}; a(t)\mathbf{x}_i, b(t)^2\mathbf{I}_d\right)\mathrm{d}t \approx \int_l^r p(t)\sum\limits_{i=1}^{N}\mathcal{N}\left(\mathbf{z}; a(t)\mathbf{x}_i, b(t)^2\mathbf{I}_d\right)\mathrm{d}t. \tag{16}$$

In the second step, we evaluate the integral by using a uniform grid of $100$ $t$ values in $[l, r]$. This two-step procedure effectively reduces computational costs while maintaining low numerical error.

**Calculation of $R(\mathbf{z}|t)$ and $R(\mathbf{z})$ (Section 4.1).** By definition,

$$R(\mathbf{z}|t) := \mathbb{E}_{\mathbf{x},\boldsymbol{\epsilon} \sim p(\mathbf{x},\boldsymbol{\epsilon}|\mathbf{z},t)}\big[r(\mathbf{x},\boldsymbol{\epsilon},t)\big] = \mathbb{E}_{\mathbf{x} \sim p(\mathbf{x}|\mathbf{z},t)}\left[c(t)\mathbf{x} + d(t)\frac{\mathbf{z} - a(t)\mathbf{x}}{b(t)}\right].$$

Notice that $p(\mathbf{x}|\mathbf{z},t) = \frac{p(\mathbf{x})}{p(\mathbf{z}|t)}p(\mathbf{z}|\mathbf{x},t)$, and $p(\mathbf{z}|\mathbf{x},t)$ is given in Eq. (13). Consequently, we have

$$R(\mathbf{z}|t) = \frac{\frac{1}{N}\sum\limits_{i=1}^{n}p(\mathbf{z}|\mathbf{x}_i,t)\left[c(t)\mathbf{x}_i + d(t)\frac{\mathbf{z}-a(t)\mathbf{x}_i}{b(t)}\right]}{\frac{1}{N}\sum\limits_{i=1}^{n}p(\mathbf{z}|\mathbf{x}_i,t)} = \frac{d(t)}{b(t)}\mathbf{z} + \left(c(t) - \frac{a(t)d(t)}{b(t)}\right)\frac{\sum\limits_{i=1}^{n}\mathcal{N}\left(\mathbf{z}; a(t)\mathbf{x}_i, b(t)^2\mathbf{I}_d\right)\mathbf{x}_i}{\sum\limits_{i=1}^{n}\mathcal{N}\left(\mathbf{z}; a(t)\mathbf{x}_i, b(t)^2\mathbf{I}_d\right)}, \tag{17}$$

which can be then explicitly calculated by scanning over all the data points $\mathbf{x}_i$.

Once we obtain $R(\mathbf{z}|t)$, using Theorem 2 and $p(t|\mathbf{z})$, $R(\mathbf{z})$ can be calculated by

$$R(\mathbf{z}) = \mathbb{E}_{t \sim p(t|\mathbf{z})}[R(\mathbf{z}|t)] = \int_0^1 p(t|\mathbf{z})R(\mathbf{z}|t)\,\mathrm{d}t. \tag{18}$$

For the integration, we utilize the selected time steps in $[l, r]$ that were used when computing $p(t|\mathbf{z})$. On another word, we ignore the parts where $p(t|\mathbf{z})$ is negligible.

**Calculation of $E(\mathbf{z})$ (Section 4.3).** $E(\mathbf{z})$ can be computed simply utilizing $p(t|\mathbf{z})$, $R(\mathbf{z}|t)$ and $R(\mathbf{z})$:

$$E(\mathbf{z}) := \mathbb{E}_{t \sim p(t|\mathbf{z})}\|R(\mathbf{z},t) - R(\mathbf{z})\|^2 = \int_0^1 p(t|\mathbf{z})\|R(\mathbf{z}|t) - R(\mathbf{z})\|^2\,\mathrm{d}t. \tag{19}$$

We again use the same time steps for estimating the integral term and reuse the terms of $p(\mathbf{z}|\mathbf{x}_i,t)$. This ensures computational efficiency while maintaining accuracy.

---

[5]Actually, this exactly matches our observation that $p(t|\mathbf{z})$ is concentrated, since $p(t|\mathbf{z}) \propto p(t)p(\mathbf{z}|t)$ for a fixed $\mathbf{z}$.

Table 4: The variance and $E$ values on the CIFAR-10 dataset. The empirical values are calculated by scanning the entire dataset, while the (theoretical) estimated values are derived from Statements 1 and 2. For reference, the values for $\|R(\mathbf{z})\|^2$ are also included to illustrate that $E(\mathbf{z})$ is significant smaller than $\|R(\mathbf{z})\|^2$. The results show that our approximation is generally accurate, except for the $E$ value in the very noisy case, where the single data point approximation becomes less accurate.

| $t_*$ | $\text{Var}_{t\sim p(t|\mathbf{z})}[t]$ | | $E(\mathbf{z})$ | | $\|R(\mathbf{z})\|^2$ |
|---|---|---|---|---|---|
| | Empirical ($\times 10^{-4}$) | Estimation ($\times 10^{-4}$) | Empirical | Estimation | Empirical |
| 0.1 | $0.0143 \pm 0.0002$ | 0.0163 | $0.558 \pm 0.005$ | 0.628 | $3894 \pm 87$ |
| 0.3 | $0.1280 \pm 0.0002$ | 0.1465 | $0.561 \pm 0.006$ | 0.628 | $3953 \pm 102$ |
| 0.5 | $0.3695 \pm 0.0004$ | 0.4069 | $0.556 \pm 0.006$ | 0.628 | $3878 \pm 108$ |
| 0.7 | $0.7008 \pm 0.0010$ | 0.7975 | $0.564 \pm 0.005$ | 0.628 | $3968 \pm 88$ |
| 0.9 | $1.3085 \pm 0.0007$ | 1.3184 | $1.822 \pm 0.245$ | 0.628 | $3310 \pm 71$ |

### A.2. Numerical Experiments

**Verification of the Single Data Point Assumption.** Recall that Statements 1 and 2 assume that the dataset contains a single data point. In this section, we conduct numerical experiments on CIFAR-10 dataset to demonstrate that this assumption provides a reasonable approximation of the variance of $p(t|\mathbf{z})$ and the error between the effective targets in practice.

For both $p(t|\mathbf{z})$ and $E(\mathbf{z})$, we calculate their values by scanning the entire dataset as shown in the previous section, and compare them with our estimated theoretical values.

For the CIFAR-10 dataset, we have $N = 50000$ and $d = 3 \times 32^2 = 3072$, from which we can derive the estimated values of $p(t|\mathbf{z})$ and $E(\mathbf{z})$ in Table 4.

As for empirical calculation, we compute the desired values via Monte Carlo sampling. Specifically, we select 5 time levels $t_* = 0.1, 0.3, 0.5, 0.7, 0.9$. For each $t_*$, we sample $M = 25$ noisy images $z_j$ by $\mathbf{z}_j = a(t_*)\mathbf{x}_{I_j} + b(t_*)\boldsymbol{\epsilon}_j, j = 1, 2, \ldots, M$. Here, $\boldsymbol{\epsilon}_j$ are independent samples from $\mathcal{N}(\mathbf{0}, \boldsymbol{I}_d)$, and $I_j$ are independent random integers from $[1, N]$. We then compute $\text{Var}_{t\sim p(t|\cdot)}[t]$, $\|R(\cdot)\|^2$ and $E(\cdot)$ for each $\mathbf{z}_j$ as we specified in Appendix A.1. Finally, we average the $M$ values to obtain the empirical values along with their statistical uncertainties. Results are shown in Table 4.

Table 4 shows that our estimations closely align with the observed values, except when $t$ gets very close to 1 (*i.e.* in highly noisy images), where the single data point approximation becomes less precise. However, even in these cases, the estimated values remain within the same order of magnitude, providing acceptable explanations for the concentration of $p(t|\mathbf{z})$ and the small error between the two effective targets.

**Visualization of $p(t|\mathbf{z})$.** We plot the value of $p(t|\mathbf{z})$ in Fig. 3 for one $\mathbf{z}$ and $t_*$ from 0.1 to 0.9. This is carried out exactly in the same manner as the variance calculation for $t$, but with AFHQ-v2 dataset at $64 \times 64$ resolution for a better visualization quality. Fig. 3 functions as a reliable visual verification of the concentration of $p(t|\mathbf{z})$.

### A.3. Evaluation of the Bound Values

In this section, we provide additional experiment details on how we compute the bound values and present the plot of the bound terms $A_i B_i$ in Fig. 4. For reference, we also include separate plots for $A_i$ and $B_i$ in Figs. 9a and 9b [6].

Recall that in Statement 3, we define $A_i$ and $B_i$ as

$$A_i = \prod_{j=i+1}^{N-1} (\kappa_i + |\eta_i|L_i), B_i = |\eta_i|\delta_i. \tag{20}$$

---

[6]An interesting fact in Fig. 9b is that, for EDM, there is a "phase change" at around $i = 50$, which is caused by a non-smooth $\delta$ value. We hypothesize that this transition occurs at a noise level high enough that the data distribution can no longer be approximated as a point distribution, leading to a noticeable shift in behavior.

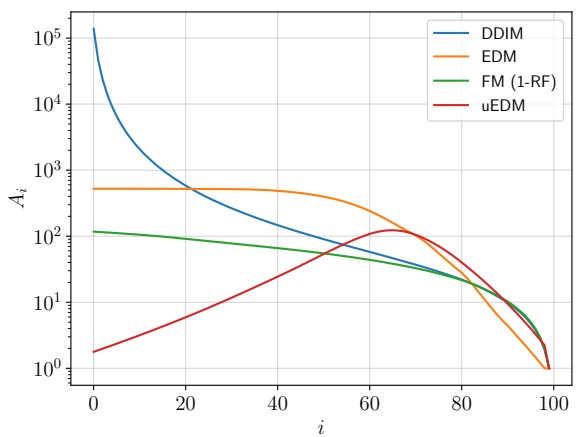

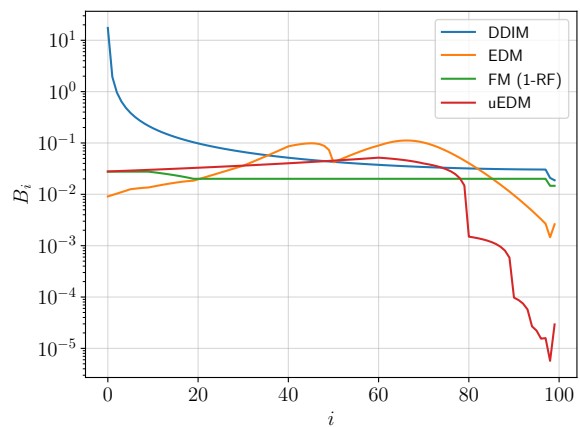

(a) The values of $A_i$ for different denoising generative models.  (b) The values of $B_i$ for different denoising generative models.

Figure 9: The bound applied on DDIM, EDM, FM and our uEDM model with a first-order ODE sampling process of $N = 100$ steps. The figures visualize the different terms $A_i$ and $B_i$ in the bound.

Since $\kappa_i$ and $\eta_i$ are already given by the configurations for each model (see Table 8), we only have to evaluate $L_i$ and $\delta_i$. Statement 3 requires the following condition to hold:

$$
\begin{cases}
\|R(\mathbf{x}_i') - R(\mathbf{x}_i'|t_i)\| \leqslant \delta_i \\
\dfrac{\|R(\mathbf{x}_i'|t_i) - R(\mathbf{x}_i|t_i)\|}{\|\mathbf{x}_i' - \mathbf{x}_i\|} \leqslant L_i
\end{cases}
\tag{21}
$$

Now we are going to pick reasonable values of $\delta_i$ and $L_i$. As mentioned in Section 4.4, it is unrealistic for this assumption to hold exactly in real data due to bad-behaviors of the effective target $R$ when the noisy image is close to pure noise or pure data. However, we aim to make the conditions hold with high probability instead of considering worst case. As a result, our choice of $\delta_i, L_i$ corresponds to *high probability case*.

**Estimation of $\delta_i$.**   We estimate $\delta_i$ using a maximum among different samples:

$$
\delta_i = \max_j \|R(\mathbf{z}_{i,j}|t_i) - R(\mathbf{z}_{i,j})\|.
\tag{22}
$$

where we sample 10 different $\mathbf{z}$ from $p(\mathbf{z}|t)$. $R(\mathbf{z}|t)$ and $R(\mathbf{z})$ values are computed as specified in Appendix A.1. We use a *maximum* value across different samples to ensure that the condition holds with high probability.

**Estimation of $L_i$.**   The condition of $L_i$ is similar to "Lipchitz constant" of $R(\cdot|t_i)$. Inspired by this, we evaluate the value of $\dfrac{\|R(\mathbf{x}_i'|t_i) - R(\mathbf{x}_i|t_i)\|}{\|\mathbf{x}_i' - \mathbf{x}_i\|}$ for $\mathbf{x}_i$ and $\mathbf{x}_i'$ that are close to each other.

To model this, we sample $\mathbf{x}_i$ from $p(\mathbf{z}|t_i)$, and let $\mathbf{x}_i' = \mathbf{x}_i + \delta\tilde{\boldsymbol{\epsilon}}$. Here, we pick $\delta = 0.01$, and $\tilde{\boldsymbol{\epsilon}} \sim \mathcal{N}(\mathbf{0}, \boldsymbol{I})$ represents a random direction, which serves as a *first-order estimation*.

Based on this, we sample 10 different pairs of $\mathbf{x}_i$ and $\mathbf{x}_i'$ for each $t_i$, and evaluate the max value of the "Lipchitz constant". In another word, we are calculating

$$
L_i = \max_j \frac{\|R(\mathbf{z}_{i,j} + \delta\tilde{\boldsymbol{\epsilon}}_j|t_i) - R(\mathbf{z}_{i,j}|t_i)\|}{\delta\|\tilde{\boldsymbol{\epsilon}}_j\|}
\tag{23}
$$

for $j = 1, 2, \ldots, 10$. Again, here we use the *maximum* value across different samples to ensure that the condition holds with high probability.

**Bound Values of the uEDM Model.** It is worth noticing that our uEDM model (see Section 5) has a non-constant weighting $w(t) \neq 1$, which doesn't match our assumption when deriving the effective target $R(\mathbf{z}|t)$ and $R(\mathbf{z})$ (as we did in Section 4.1). However, we choose not to introduce more mathematical complexities, and instead use the formulas above to calculate the bound value for uEDM. This implies that the bound for uEDM is no longer mathematically strict, but it can still serve as a reasonable intuition for the choice of our uEDM configuration.

**Absolute Magnitude of the Bound Values.** As shown in Fig. 4, one might observe that the magnitude of the bound (around $10^2$ to $10^6$) is actually significantly larger compared to the typical magnitude of $\|\mathbf{x}_N\|$ (which is around $\sigma_\mathrm{d}\sqrt{d} \sim 10^1$). We hypothesize that this is because of the following two reasons:

(1) We are assuming the error accumulating on each step, which might not be the case in practice, as studied in our discussion of SDE samplers in Section 6.2.

(2) When the noisy image approaches clean data, some properties of the effective target $R$ become bad (*e.g.* the Lipchitz constant $L_i$ will be very big). This leads to a large error estimation, but in real cases, as the neural network will smooth the learned function, the error is typically smaller. If we consider ignoring the last 10 steps in our bound value, the bound will be in a reasonable range (approximately 10 for FM and uEDM, 140 for EDM, and $> 10^5$ for DDIM).

# B. Additional Experimental Details

### B.1. General Experiment Configurations

We implement our main code base using Google TPU and the JAX (Bradbury et al., 2018) platform, and run most of our experiments on TPU v2 and v3 cores. As the official codes are mostly provided as GPU code, we re-implemented most of the previous work in JAX. For the faithfulness of our re-implementation, please refer to Appendix B.3.

**FID Evaluation.** For evaluation of the generative models, we calculate FID (Heusel et al., 2017) between 50,000 generated images and all available real images without any augmentation. We used the pre-trained Inception-v3 model provided by StyleGAN3 (Karras et al., 2021) and converted it into a model class compatible with JAX. As we have reproduced most of the results in Appendix B.3, we believe that our FID calculation is reliable.

**Noise Conditioning Removal.** When we refer to "removing noise conditioning", technically we set the scalar before passing into the time-embedding to zero. Alternatively, we can also set the embedded time vector to zero. The results turn out to have negligible differences.

**iDDPM (x-pred).** We design a $\mathbf{x}$-prediction version of iDDPM to show the generalizability of our theoretical framework. During training time, we simply change the target $r(\mathbf{x}, \boldsymbol{\epsilon}, t)$ or $r(\mathbf{x}, \boldsymbol{\epsilon})$ to be $\mathbf{x}$. The sampling algorithm has to be modified accordingly, and we directly translate the $\mathbf{x}$-prediction to $\boldsymbol{\epsilon}$-prediction by Eq. (1).

**ADM.** In the original work of ADM, Dhariwal & Nichol (2021) don't provide result on the CIFAR-10 dataset in class-unconditional settings. In our implementation of ADM, we keep the main method of learning $\boldsymbol{\epsilon}$-prediction and the variance $\boldsymbol{\Sigma}$ simultaneously, but employ it on the class-unconditional CIFAR-10 task. Notice that this ADM formulation is also *not* included in our theoretical framework, but it still gives a reasonable result after removing noise conditioning (see Table 2).

**Hyperparameters.** A table of selected important hyperparameters can be found in Table 5. For ICM and ECM we use the RAdam (Liu et al., 2020) optimizer, while for all other models we use the Adam (Kingma & Ba, 2015) optimizer. Also, we set the parameter $\beta_2$ to 0.95 to stabilize the training process.

For all CIFAR-10 experiments, we used the architecture of NCSN++ in Song et al. (2021b), with 56M parameters. For the ImageNet 32×32 experiment, we used the same architecture but a larger scale, with a total of 210M parameters. For experiments on classifier-free guidance on ImageNet 256×256, we strictly follow the SiT-B/2 model with 400k training steps in Ma et al. (2024).

We highlight that for all experiments, we only tune hyperparameters on the noise-conditional model, and then *directly use exactly the same hyperparameters* for the noise-unconditional model and don't perform any further hyperparameter tuning. Thus, we expect that tuning these hyperparameters may further improve the performance of the noise-unconditional model.

**Class-Conditional Generation on CIFAR-10.** For the class-conditional CIFAR-10 experiments, we use exactly the same configurations and hyperparameters of EDM and FM with the unconditional generation case, except that we train the network with labels. For the conditioning on labels, we use the architecture in Karras et al. (2022). We do not apply any kind of guidance at inference time.

**FFHQ** $64\times64$ **Experiments.** For FFHQ $64\times64$ experiments, we directly use the code provided by Karras et al. (2022) and run it on 8 H100 GPUs. We keep all hyperparameters the same as the original code in the experiments. For the removal of noise-conditioning, we simply set the $c_\text{noise}$ variable in the code to zero.

Table 5: Selected important hyperparameters in our main experiments.

| Experiment | Duration | Warmup Epochs | Batch Size | Learning Rate | EMA Schedule | EMA Half-life Images | Dropout |
|---|---|---|---|---|---|---|---|
| iDDPM & ADM | 100M | 200 | 2048 | $8\times10^{-4}$ | EDM | 50M | 0.15 |
| iDDPM($\mathbf{x}$-pred) | 200M | 200 | 2048 | $1.2\times10^{-3}$ | EDM | 50M | 0.15 |
| DDIM | 100M | 200 | 512 | $4\times10^{-4}$ | EDM | 50M | 0.1 |
| FM | 100M | 200 | 2048 | $8\times10^{-4}$ | EDM | 50M | 0.15 |
| FM ImageNet32 | 256M | 64 | 2048 | $2\times10^{-4}$ | EDM | 50M | 0 |
| EDM | 200M | 200 | 512 | $1\times10^{-3}$ | EDM | 0.5M | 0.13 |
| uEDM (ours) | 200M | 200 | 512 | $4\times10^{-4}$ | EDM | 0.5M | 0.2 |
| ICM & ECM | 400M | 0 | 1024 | $1\times10^{-4}$ | Const(0.99993) | - | 0.3 |
| SiT-B/2 | 100M | 0 | 256 | $1\times10^{-4}$ | Const(0.9999) | - | 0 |

## B.2. Special Experiments

This section covers the specific experiment details in Section 6.2.

**DDIM-iDDPM Interpolate Sampler.** In the analysis of stochasticity, we examine VP diffusion models with cosine and linear $\bar{\alpha}(t)$ schedule with and without noise conditioning, using a customized interpolate sampler featured by $\lambda$, which is given by Eq. (56). For the cosine schedule, we use 500 sampling steps, while for the linear schedule, we use 100 sampling steps. As discussed in Appendix D.2, when $\lambda = 1$, the model with the cosine schedule has the same setting as "iDDPM" in Table 2; when $\lambda = 0$, the model with the linear schedule has the same setting as "DDIM ODE 100" in Table 2.

Results for the experiment are shown in Table 6, and the result for the cosine schedule model is visualized in Fig. 6. From Table 6 one can also find a consistent trend that as $\lambda$ increase, the degradation of FID becomes smaller for the noise-unconditional model, regardless of the specific schedule of $\bar{\alpha}(t)$.

**Alternative Architectures: Noise Level Predictor and Noise-like Condition.** In our experiment of Noise Level Predictor, we train a very lightweight network to predict the noise level $t$ given the input $\mathbf{z}$. To be specific, our predictor network only contains two convolutional layers with relu activation, followed by a global average pooling layer and a linear layer. The network has no more than 30K parameters, so it hardly affects the expressiveness of the whole model.

It's worth noticing that directly predicting the input $t$ is usually not desirable, as the value of $t$ may have different ranges for different models. Instead, for each specific model we choose a customized target for the prediction.

Training objectives for different models are shown below ($P_\phi$ represents the predictor network):

- FM: $\mathcal{L} = \mathbb{E}_{\mathbf{x}\sim p_\text{data}, \boldsymbol{\epsilon}\sim\mathcal{N}(\mathbf{0},\boldsymbol{I})}\mathbb{E}_{t\sim\mathcal{U}[0,1]}\left[P_\phi((1-t)\mathbf{x}+t\boldsymbol{\epsilon})-t\right]^2.$

- VP models: $\mathcal{L} = \mathbb{E}_{\mathbf{x}\sim p_\text{data}, \boldsymbol{\epsilon}\sim\mathcal{N}(\mathbf{0},\boldsymbol{I})}\mathbb{E}_{t\sim\mathcal{U}[0,1]}\left[P_\phi(\sqrt{1-t}\mathbf{x}+\sqrt{t}\boldsymbol{\epsilon})-t\right]^2.$

- EDM models: $\mathcal{L} = \mathbb{E}_{\mathbf{x}\sim p_\text{data}, \boldsymbol{\epsilon}\sim\mathcal{N}(\mathbf{0},\boldsymbol{I})}\mathbb{E}_{t\sim\exp\mathcal{N}(-1.2,1.2^2)}\left[\frac{1}{P_\phi(c_\text{in}(t)(\mathbf{x}+t\boldsymbol{\epsilon}))+1}-\frac{1}{1+t}\right]^2.$

Here, for EDM, we apply a transformation $y \rightarrow \frac{1}{1+y}$, mapping the original noise level $t \in (0, +\infty)$ in EDM to $[0, 1]$.

Table 6: Comparison of inference performance between noise-conditional and noise-unconditional models. The *left* panel uses a cosine noise schedule, while the *right* panel uses a linear noise schedule. Both panels compare the performance of noise-conditional and noise-unconditional settings across different values of the ablation sampler coefficient $\lambda$, ranging from 0.0 to 1.0.

| | cosine schedule | | linear schedule | |
|---|---|---|---|---|
| $\lambda$ | FID | Change | FID | Change |
| 0.0 | $2.98 \rightarrow 34.56$ | +31.58 | $\mathbf{3.99} \rightarrow 40.90$ | +36.91 |
| 0.2 | $2.60 \rightarrow 30.34$ | +27.74 | $4.09 \rightarrow 36.04$ | +31.95 |
| 0.4 | $\mathbf{2.52} \rightarrow 21.33$ | +18.81 | $4.45 \rightarrow 28.08$ | +23.63 |
| 0.6 | $2.59 \rightarrow 12.47$ | + 9.88 | $4.95 \rightarrow 18.32$ | +13.37 |
| 0.8 | $2.77 \rightarrow \phantom{0}7.53$ | + 4.76 | $5.90 \rightarrow 10.36$ | + 4.46 |
| 1.0 | $3.13 \rightarrow \phantom{0}\mathbf{5.51}$ | + 2.38 | $8.07 \rightarrow \mathbf{10.85}$ | + 2.78 |

Experimentally, the MSE loss for all these settings all have a magnitude on the order of $10^{-4}$, which means that even our very lightweight predictor can predict the noise level with a mean error of about $0.01$.

In our Noise-like Conditioning experiment, the same lightweight network as mentioned above is connected to a noise-conditional U-Net, as visualized in Fig. 7c. This joint training architecture is noise-unconditional. The main difference between this experiment and the "Noise Level Predictor" experiment is that there is *no supervision* on the intermediate output, so it may not be the noise level $t$ itself.

For the experiments in Fig. 7 column (b) and (c), we again use the same set of hyperparameters and configurations as the noise-conditional and noise-unconditional experiments (in columns (a) and (d)), to ensure a fair comparison. However, a subtle detail is that in the implementation of iDDPM in Fig. 7, the results for noise-conditional and noise-unconditional experiments are not the same as the results in Table 2. This is due to that we use $T = 500$ instead of $T = 4000$ for the training process.

### B.3. Our Reimplementation Faithfulness

As we have mentioned, we reimplement most of the models in the platform of JAX and TPU. Table 7 shows the comparison of our reimplementation and the original reported results.

For some models, the reproduction doesn't meet our expectations. For example, since the official code for iCT is currently not open-sourced, we can only follow all configurations mentioned in their work, and get a best FID score of 2.59 (compared with the originally reported score of 2.46). For EDM on FFHQ-64, we directly run the given official code with the VP configuration on 8 H100 GPUs, but still can't reproduce the result. We suspect that the difference may come from random variance or different package versions used in the experiments.

Note that iDDPM (Nichol & Dhariwal, 2021) only reports results with 1000 and 4000 training steps. Due to computational constraints, we only evaluate the model with 500 sampling steps. Since our 500-step result has already been better than the 1000-step result reported in their work, we believe that we successfully reproduced the model.

After all, our goal is to compare the performance of noise-conditional and noise-unconditional models, and the absolute performance is not the focus of our work. Thus, even though we haven't fully reproduced some of the results, we believe that our comparison is still meaningful.

## C. Supplementary Theoretical Details

### C.1. Proof of the Effective Target

**Theorem 1.** *The original regression loss function with $t$ condition shown in Eq. (2) with $w(t) = 1$*

$$\mathcal{L}(\boldsymbol{\theta}) = \mathbb{E}_{\mathbf{x}, \boldsymbol{\epsilon}, t}\left[\left\| NN_{\boldsymbol{\theta}}(\mathbf{z}|t) - r(\mathbf{x}, \boldsymbol{\epsilon}, t)\right\|^2\right]$$

*is equivalent to the loss function with the effective target shown in Eq. (5)*

$$\mathcal{L}'(\boldsymbol{\theta}) = \mathbb{E}_{\mathbf{z} \sim p(\mathbf{z}), t \sim p(t|\mathbf{z})}\left[\left\| NN_{\boldsymbol{\theta}}(\mathbf{z}|t) - R(\mathbf{z}|t)\right\|^2\right]$$

Table 7: Comparison of our reimplementation and the original reported results.

| Model | Sampler | NFE | FID Reproduced | Original |
|---|---|---|---|---|
| **CIFAR-10** | | | | |
| iDDPM (Nichol & Dhariwal, 2021) | - | 500 | 3.13 | - |
| | | 1000 | - | 3.29 |
| | | 4000 | - | 2.90 |
| DDIM (Song et al., 2021a) | ODE | 100 | 3.99 | 4.16 |
| | ODE | 1000 | 2.85 | 4.04 |
| EDM (Karras et al., 2022) | Heun | 35 | 1.99 | 1.97 |
| 1-RF (Liu et al., 2023) | RK45 | ~127 | 2.53 | 2.58 |
| iCT (Song & Dhariwal, 2024) | - | 2 | 2.59 | 2.46 |
| **CIFAR-10 Class-conditional** | | | | |
| EDM (Karras et al., 2022) | Heun | 35 | 1.79 | 1.76 |
| **ImageNet 32×32** | | | | |
| FM (Lipman et al., 2023) | Euler | 100 | 5.15 | - |
| FM (Lipman et al., 2023) | RK45 | ~125 | 4.30 | 5.02 |
| **FFHQ 64×64** | | | | |
| EDM (Karras et al., 2022) | Heun | 79 | 2.64 | 2.39 |

only up to a constant term that is independent of $\boldsymbol{\theta}$, where

$$R(\mathbf{z}|t) = \mathbb{E}_{(\mathbf{x},\boldsymbol{\epsilon}) \sim p(\mathbf{x},\boldsymbol{\epsilon}|\mathbf{z},t)}\big[r(\mathbf{x},\boldsymbol{\epsilon},t)\big].$$

Here, $p(\mathbf{z})$ is the marginalized distribution of $\mathbf{z} := a(t)\mathbf{x} + b(t)\boldsymbol{\epsilon}$ in Eq. (1), under the joint distribution $p(\mathbf{x},\boldsymbol{\epsilon},t) := p(\mathbf{x})p(\boldsymbol{\epsilon})p(t)$.

*Proof.* The original regression loss function with $t$ condition can be rewritten as

$$\begin{aligned}
\mathcal{L}(\boldsymbol{\theta}) &= \mathbb{E}_{\mathbf{z} \sim p(\mathbf{z}), t \sim p(t|\mathbf{z})} \mathbb{E}_{(\mathbf{x},\boldsymbol{\epsilon}) \sim p(\mathbf{x},\boldsymbol{\epsilon}|\mathbf{z},t)} \Big[ \big\| \mathrm{NN}_{\boldsymbol{\theta}}(\mathbf{z}|t) - r(\mathbf{x},\boldsymbol{\epsilon},t) \big\|^2 \Big] \\
&= \mathbb{E}_{\mathbf{z} \sim p(\mathbf{z}), t \sim p(t|\mathbf{z})} \Big[ \big\| \mathrm{NN}_{\boldsymbol{\theta}}(\mathbf{z}|t) - \mathbb{E}_{(\mathbf{x},\boldsymbol{\epsilon}) \sim p(\mathbf{x},\boldsymbol{\epsilon}|\mathbf{z},t)}\big[r(\mathbf{x},\boldsymbol{\epsilon},t)\big] \big\|^2 + \mathbb{V}_{(\mathbf{x},\boldsymbol{\epsilon}) \sim p(\mathbf{x},\boldsymbol{\epsilon}|\mathbf{z},t)}\big[r(\mathbf{x},\boldsymbol{\epsilon},t)\big] \Big] \\
&= \mathbb{E}_{\mathbf{z} \sim p(\mathbf{z}), t \sim p(t|\mathbf{z})} \Big[ \big\| \mathrm{NN}_{\boldsymbol{\theta}}(\mathbf{z}|t) - R(\mathbf{z}|t) \big\|^2 + const \Big] \\
&= \mathbb{E}_{\mathbf{z} \sim p(\mathbf{z}), t \sim p(t|\mathbf{z})} \Big[ \big\| \mathrm{NN}_{\boldsymbol{\theta}}(\mathbf{z}|t) - R(\mathbf{z}|t) \big\|^2 \Big] + const = \mathcal{L}'(\boldsymbol{\theta}) + const.
\end{aligned} \quad (24)$$

This finishes the proof. □

**Theorem 2.** *The original regression loss function without noise conditioning*

$$\mathcal{L}(\boldsymbol{\theta}) = \mathbb{E}_{\mathbf{x},\boldsymbol{\epsilon},t}\Big[ \big\| NN_{\boldsymbol{\theta}}(\mathbf{z}) - r(\mathbf{x},\boldsymbol{\epsilon},t) \big\|^2 \Big]$$

*is equivalent to the loss function with the effective target shown in Eq. (7)*

$$\mathcal{L}'(\boldsymbol{\theta}) = \mathbb{E}_{\mathbf{z} \sim p(\mathbf{z})}\Big[ \big\| NN_{\boldsymbol{\theta}}(\mathbf{z}) - R(\mathbf{z}) \big\|^2 \Big]$$

*only up to a constant term that is independent of $\boldsymbol{\theta}$, where*

$$R(\mathbf{z}) = \mathbb{E}_{t \sim p(t|\mathbf{z})}\big[R(\mathbf{z}|t)\big].$$

*Defintions on $p(\mathbf{z})$ and $p(t|\mathbf{z})$ are the same as in Theorem 1.*

*Proof.* The original regression loss function without noise conditioning can be rewritten as

$$
\begin{aligned}
\mathcal{L}(\boldsymbol{\theta}) &= \mathbb{E}_{\mathbf{z}\sim p(\mathbf{z})}\mathbb{E}_{(\mathbf{x},\boldsymbol{\epsilon},t)\sim p(\mathbf{x},\boldsymbol{\epsilon},t|\mathbf{z})}\Big[\big\|\mathrm{NN}_{\boldsymbol{\theta}}(\mathbf{z}) - r(\mathbf{x},\boldsymbol{\epsilon},t)\big\|^2\Big] \\
&= \mathbb{E}_{\mathbf{z}\sim p(\mathbf{z})}\Big[\big\|\mathrm{NN}_{\boldsymbol{\theta}}(\mathbf{z}) - \mathbb{E}_{(\mathbf{x},\boldsymbol{\epsilon},t)\sim p(\mathbf{x},\boldsymbol{\epsilon},t|\mathbf{z})}\big[r(\mathbf{x},\boldsymbol{\epsilon},t)\big]\big\|^2 + \mathbb{V}_{(\mathbf{x},\boldsymbol{\epsilon},t)\sim p(\mathbf{x},\boldsymbol{\epsilon},t|\mathbf{z})}\big[r(\mathbf{x},\boldsymbol{\epsilon},t)\big]\Big] \\
&= \mathbb{E}_{\mathbf{z}\sim p(\mathbf{z}),t\sim p(t|\mathbf{z})}\Big[\big\|\mathrm{NN}_{\boldsymbol{\theta}}(\mathbf{z}) - \mathbb{E}_{(\mathbf{x},\boldsymbol{\epsilon},t)\sim p(\mathbf{x},\boldsymbol{\epsilon},t|\mathbf{z})}\big[r(\mathbf{x},\boldsymbol{\epsilon},t)\big]\big\|^2 + const\Big] \\
&= \mathbb{E}_{\mathbf{z}\sim p(\mathbf{z}),t\sim p(t|\mathbf{z})}\Big[\big\|\mathrm{NN}_{\boldsymbol{\theta}}(\mathbf{z}) - \mathbb{E}_{(\mathbf{x},\boldsymbol{\epsilon},t)\sim p(\mathbf{x},\boldsymbol{\epsilon},t|\mathbf{z})}\big[r(\mathbf{x},\boldsymbol{\epsilon},t)\big]\big\|^2\Big] + const
\end{aligned}
\tag{25}
$$

And notice that

$$
\mathbb{E}_{(\mathbf{x},\boldsymbol{\epsilon},t)\sim p(\mathbf{x},\boldsymbol{\epsilon},t|\mathbf{z})}\big[r(\mathbf{x},\boldsymbol{\epsilon},t)\big] = \mathbb{E}_{t\sim p(t|\mathbf{z})}\mathbb{E}_{(\mathbf{x},\boldsymbol{\epsilon})\sim p(\mathbf{x},\boldsymbol{\epsilon}|\mathbf{z},t)}\big[r(\mathbf{x},\boldsymbol{\epsilon},t)\big] = \mathbb{E}_{t\sim p(t|\mathbf{z})}R(\mathbf{z}|t).
\tag{26}
$$

So

$$
\begin{aligned}
\mathcal{L}(\boldsymbol{\theta}) &= \mathbb{E}_{\mathbf{z}\sim p(\mathbf{z}),t\sim p(t|\mathbf{z})}\Big[\big\|\mathrm{NN}_{\boldsymbol{\theta}}(\mathbf{z}) - \mathbb{E}_{(\mathbf{x},\boldsymbol{\epsilon},t)\sim p(\mathbf{x},\boldsymbol{\epsilon},t|\mathbf{z})}\big[r(\mathbf{x},\boldsymbol{\epsilon},t)\big]\big\|^2\Big] + const \\
&= \mathbb{E}_{\mathbf{z}\sim p(\mathbf{z}),t\sim p(t|\mathbf{z})}\Big[\big\|\mathrm{NN}_{\boldsymbol{\theta}}(\mathbf{z}) - R(\mathbf{z})\big\|^2\Big] + const = \mathcal{L}'(\boldsymbol{\theta}) + const.
\end{aligned}
$$

This finishes the proof. $\qquad\square$

## C.2. Approximation of the Variance of $p(t|\mathbf{z})$

We claim in Statement 1 that, for a fixed noisy image $\mathbf{z}$ whose true noise level is $t_*$, the posterior variance of $p(t|\mathbf{z})$ scales like $t_*^2/2d$. In this section, we first derive a $\mathcal{O}(t_*^2/d)$ upper bound on the variance under minimal technical assumptions. While obtaining the exact constant requires delicate optimizations, our Big-O presentation keeps the proof accessible. We then present a concise, intuitive argument to recover the $t_*^2/2d$ scaling, which— as confirmed by our numerical results in Appendix A.2—serves as an accurate and practical estimate of $\mathrm{Var}_{p(t|\mathbf{z})}[t]$.

### C.2.1. RIGOROUS UPPER BOUND ON VARIANCE.

**Single Data Point Case.** We first consider the case where $N = 1$. For brevity, write $\mathbf{x} = \mathbf{x}_1$. Recall that our goal is to bound, *with high probability* over the randomness in $z \sim p(z)$, the posterior variance $\mathrm{Var}_{t|\mathbf{z}}[t]$.

Note that some ill-behaved $\mathbf{z}$ can lead to strange distribution of $t$. Thus we work under the high-probability regime in which $\mathbf{z}$ concentrates around its typical behavior.

Because $\boldsymbol{\epsilon} \sim \mathcal{N}(\mathbf{0}, \boldsymbol{I}_d)$ is rotationally invariant, we may, without loss of generality, assume $\mathbf{x}$ to only have 1 nonzero coordinate.

**Theorem 3.** *Assume that*

$$
\mathbf{x} = (x, 0, 0, \cdots, 0), x \in \left[-\sqrt{d}, \sqrt{d}\right]
$$
$$
\mathbf{z} = (1 - t_0)\mathbf{x} + t_0\boldsymbol{\epsilon}, \boldsymbol{\epsilon} = (\epsilon_0, \boldsymbol{\epsilon}')
$$

*Here, $\epsilon_0$ is a scalar that is the first coordinate of $\boldsymbol{\epsilon}$, and $\boldsymbol{\epsilon}'$ is its other coordinates.*

*We further assume that*

$$
t_0 \in \left[\frac{1}{d}, 1 - \frac{1}{d}\right], \quad \big|\|\boldsymbol{\epsilon}'^2| - d\big| = \mathcal{O}(\sqrt{d}\log d), \quad |\epsilon_0|^2 \leqslant \log d
\tag{27}
$$

*Then, we have $\mathrm{Var}_{t|\mathbf{z}}(t) = \mathcal{O}(t_0^2/d)$.*

Note that Eq. (27) occur with probability $1 - \mathcal{O}(\frac{1}{d})$, due to the norm-concentration properties of the Gaussian distribution.

*Proof.* We directly compute the probability density functions as follows.

Firstly, we have

$$p(t|\mathbf{z}) \propto p(\mathbf{z}|t)p(t) \propto \frac{1}{t^d} \exp\left(-\frac{1}{2}\left(\frac{|\mathbf{z} - (1-t)\mathbf{x}|}{t}\right)^2\right)$$

since the distribution $p(\mathbf{z}|t)$ is simply a Gaussian with mean $(1-t)\mathbf{x}$ and variance $t^2$ per dimension.

Looking into the exponent, it can be simplified as

$$\left(\frac{|\mathbf{z} - (1-t)\mathbf{x}|}{t}\right)^2 = \left(\frac{|(t-t_0)\mathbf{x} + t_0\boldsymbol{\epsilon}|}{t}\right)^2 = \left|\left(1 - \frac{t_0}{t}\right)\mathbf{x} + \frac{t_0}{t}\mathbf{z}\right|^2$$

$$= \left(\left(1 - \frac{t_0}{t}\right)x + \frac{t_0}{t}\epsilon_0\right)^2 + \left(\frac{t_0}{t}\right)^2 |\boldsymbol{\epsilon}'|^2$$

which is a quadratic function on $\frac{t_0}{t}$. We write it as $a\left(\frac{t_0}{t} - \mu\right)^2 + C$ for some constants $a, \mu, C$ independent of $t$, where

$$a = (x - \epsilon_0)^2 + |\boldsymbol{\epsilon}'|^2 = x^2 - x\mathcal{O}(\sqrt{\log d}) + d + \mathcal{O}(\sqrt{d}\log d)$$
$$= x^2 + d + \mathcal{O}(\sqrt{d}\log d) \tag{28}$$

and

$$a\mu = x(x - \epsilon_0) = x^2 - \mathcal{O}(x\sqrt{\log d}) \tag{29}$$

by the assumption Eq. (27).

Substituting back, we get

$$p(t|\mathbf{z}) \propto \frac{1}{t^d} \exp\left(-\frac{1}{2}a\left(\frac{t_0}{t} - \mu\right)^2\right)$$

We find its maximum by differentiating its log-density:

$$\frac{\partial \log p(t|\mathbf{z})}{\partial t} = -\frac{d}{t} + a\left(\frac{t_0}{t} - \mu\right)\frac{t_0}{t^2} = \frac{1}{t}\left(a\left(\frac{t_0}{t}\right)^2 - a\mu\frac{t_0}{t} - d\right)$$

Let $\lambda_1 > 0, \lambda_2 < 0$ be the two roots of the equation $f(X) := aX^2 - a\mu X - d = 0$ (since $a$ and $d$ are both positive).
Notice that $\lambda_1\lambda_2 = -\frac{d}{a}$. We claim that

$$\lambda_1 = 1 + \mathcal{O}\left(\frac{\log d}{\sqrt{d}}\right), \lambda_2 = -\frac{d}{a}\left(1 + \mathcal{O}\left(\frac{\log d}{d}\right)\right) = -\Theta(1) \tag{30}$$

This is because

$$f(1) = \mathcal{O}(\sqrt{d}\log d), f'(1) = x^2 + 2d + \mathcal{O}(\sqrt{d}\log d)$$

by Eq. (28) and Eq. (30), from which we derive the desired root bounds.

Using the factorization into roots, we have

$$\frac{\partial \log p(t|\mathbf{z})}{\partial t} = \frac{1}{t}\left(a\left(\frac{t_0}{t}\right)^2 - a\mu\frac{t_0}{t} - d\right) = \frac{1}{t}a\left(\frac{t_0}{t} - \lambda_1\right)\left(\frac{t_0}{t} - \lambda_2\right)$$

$$= \frac{1}{t}\left(\frac{t_0}{t} - \lambda_1\right)\Omega(d)$$

by Eq. (28) and Eq. (30).

Denote $t^* = \dfrac{t_0}{\lambda_1}$ as the unique maximizer of $\log p(t|\mathbf{z})$. As a first step, we show that in range $(0, 3t^*]$, the probability density of $p(t|\mathbf{z})$ is more concentrated than a Gaussian with variance $\mathcal{O}(t_0^2/d)$. Intuitively, it is due to the fact that in this range,

$$\frac{\partial \log p(t|\mathbf{z})}{\partial t} = \frac{1}{t}\left(\frac{t_0}{t} - \frac{t_0}{t^*}\right)\Omega(d) = \frac{t_0(t^* - t)}{t^2 t^*}\Omega(d) = (t^* - t)\Omega\left(\frac{d}{t_0^2}\right)$$

utilizing Eq. (30). This inequality implies that in this range, the gradient of this probability density is sharper than $\mathcal{N}(t^*, \mathcal{O}(t_0^2/d))$.

To be specific, when $0 < t < t' \leqslant t^*$ or $t^* \leqslant t' < t \leqslant 3t^*$, we have

$$\log p(t'|\mathbf{z}) - \log p(t|\mathbf{z}) = \int\limits_t^{t'} \frac{\partial \log p(t|\mathbf{z})}{\partial t}\mathrm{d}t = \Omega\left(\frac{d}{t_0^2}\right) \cdot \left((t - t^*)^2 - (t' - t^*)^2\right) \tag{31}$$

where the RHS is exactly the difference in log-probability density of $\mathcal{N}(t^*, \mathcal{O}(t_0^2/d))$ at $t$ and $t'$. This fact supports that in this range, $p(t|\mathbf{z})$ is sharper than $\mathcal{N}(t^*, \mathcal{O}(t_0^2/d))$, or

$$\mathbb{E}_{t \sim p'}[(t - t^*)^2] \leqslant \mathbb{E}_{t \sim q}[(t - t^*)^2] = \mathcal{O}(t_0^2/d)$$

where $p', q$ denotes the distribution of $p(t|\mathbf{z})$ and $\mathcal{N}(t^*, \mathcal{O}(t_0^2/d))$ restricted on $(0, 3t^*]$, respectively.

Finally, we only need to consider the part when $3t^* < t \leqslant 1$, where we are going to prove that

$$\Pr_{t \sim p(t|\mathbf{z})}[t > 3t^*]$$

is small. In this case, according to Eq. (31), for any $t' \in [t^*, 2t^*]$:

$$\frac{p(t|\mathbf{z})}{p(t'|\mathbf{z})} \leqslant \exp\left(-\Omega\left(\frac{d}{t_0^2}\right) \cdot (4t^{*2} - t^{*2})\right) = \exp(-\Omega(d))$$

which is actually exponentially small. This implies that

$$\Pr_{t \sim p(t|\mathbf{z})}[t > 3t^*] \leqslant \exp(-\Omega(d)) \cdot \min_{t' \in [t^*, 2t^*]} p(t'|\mathbf{z}) \leqslant \exp(-\Omega(d)) \cdot \frac{1}{t^*} = \mathcal{O}\left(\frac{1}{d^3}\right)$$

Consequently, the contribution of this tail to the variance is $\mathcal{O}(1/d^3)$, which is negligible compared to $\mathcal{O}(t_0^2/d)$. Combining the two regimes, we conclude that

$$\mathrm{Var}_{p(t|\mathbf{z})}[t] \leqslant \mathbb{E}_{t \sim p(t|\mathbf{z})}[(t - t^*)^2] = \mathcal{O}(t_0^2/d)$$

$\square$

**Multiple Data Points Case.** We now turn to the setting of $N > 1$ data points. Intuitively, one can identify the ground-truth clean image that generates the given noisy image, as long as the noise is not too large to swamp all signal. This can be done simply by comparing inner product: the noisy observation $\mathbf{z}$ will correlate most strongly with its corresponding clean sample $\mathbf{x}_i$, and only weakly with all others.

**Lemma 1.** *Suppose that $\mathbf{x}_1, \mathbf{x}_2, \ldots, \mathbf{x}_N$ are i.i.d Gaussian samples from $\mathcal{N}(\mathbf{0}, \mathbf{I}_d)$, and $\mathbf{z} = (1 - t_0)\mathbf{x}_i + t\boldsymbol{\epsilon}$ for some $i$.*

*Then, we have the following two properties hold with probability $1 - \frac{1}{d}$:*

$$\begin{cases} |\langle \mathbf{x}_j, \mathbf{z}\rangle| & = \mathcal{O}(\sqrt{d}(\log N + \log d)), \qquad \forall j \neq i \\ |\langle \mathbf{x}_i, \mathbf{z}\rangle - (1 - t_0)|\mathbf{x}_i|^2| & = \mathcal{O}(\sqrt{d \log d}) \end{cases}$$

*Proof.* We first deal with the second inequality. We have

$$\langle \mathbf{x}_i, \mathbf{z} \rangle - (1 - t_0)|\mathbf{x}_i|^2 = \langle \mathbf{x}_i, (1 - t_0)\mathbf{x}_i + t\boldsymbol{\epsilon} \rangle - (1 - t_0)|\mathbf{x}_i|^2 = t_0 \langle \mathbf{x}_i, \boldsymbol{\epsilon} \rangle \sim \mathcal{N}(\mathbf{0}, |\mathbf{x}_i|I_d)$$

Using standard Gaussian tail bounds, this probability that $| \langle \mathbf{x}_i, \mathbf{z} \rangle - (1 - t_0)|\mathbf{x}_i|^2| \geqslant k\sqrt{d \log d}$ is at most

$$\exp\left(-\frac{k^2 d \log d}{|\mathbf{x}_i|^2}\right) = \exp(-k^2 \log d) = d^{-\Omega(k^2)}$$

Therefore, there exists a constant $k$ such that the probability above is at most $\frac{1}{dN}$.

For the first inequality, notice that

$$\langle \mathbf{x}_j, \mathbf{z} \rangle = (1 - t) \langle \mathbf{x}_i, \mathbf{x}_j \rangle + t \langle \mathbf{x}_j, \boldsymbol{\epsilon} \rangle$$

each term is the dot product of two normal Gaussian variables, which is still sub-exponential.

In this way, we derive that each term is at most $\mathcal{O}(d(\log N + \log d))$ with probability $1 - \frac{1}{dN}$, so union bound over all the $i$ gives the desired result. $\qquad\square$

**Theorem 4.** *There exists a constant $C$ such that when $1 - t_0 \geqslant C\sqrt{\dfrac{\log N + \log d}{d}}$, we can recover the correct $i$ with probability at least $1 - \dfrac{1}{d}$.*

*Proof.* Using Lemma 1, we can compute the dot products of all $\mathbf{x}_i$ with $\mathbf{z}$ and take the largest one. $\qquad\square$

This theorem implies that the concentration property of $p(t|\mathbf{z})$ can be extended to multiple data points case, even when there are *exponentially-many* data points. By first recovering the correct clean image, the multi–data–point setting reduces immediately to the single–point analysis, and hence the posterior variance bound $\mathcal{O}(t_0^2/d)$ extends to the case of exponentially many candidates.

### C.2.2. INTUITIVE DERIVATION OF THE APPROXIMATE CONSTANT

In this section, we show the reason why we estimate the variance as $t_0^2/2d$.

Firstly, as we have seen in Theorem 4, the ground-truth clean image can be recovered with high confidence, even with a prior of exponentially-many data points. As a result, we revert to the single data point setting.

For simplicity, we restrict attention to the bulk regime $t \in \left[\dfrac{1}{d}, 1 - \dfrac{1}{d}\right]$, postponing edge–case analysis to future work.

Now, we can still view the data point $\mathbf{x}$ as $(x, 0, \ldots, 0)$ due to symmetry, where $x$ stands for the norm of $\mathbf{x}$. Because the dominant contribution to the likelihood comes from the $d - 1$ noise dimensions when $d$ is large, we further approximate by neglecting the signal component in the first coordinate (*i.e.*, assume $\mathbf{x} = \mathbf{0}$).

On this assumption, we have

$$\mathbf{z}' = (1 - t)\mathbf{x}' + t\boldsymbol{\epsilon}' = t\boldsymbol{\epsilon}.$$

and denote $|\mathbf{z}| = t_0\sqrt{d}$. We have

$$p(t|\mathbf{z}) \propto \frac{1}{t^d} \exp\left(-\frac{dt_0^2}{2t^2}\right)$$

The variance of this distribution can be calculated exactly from integration, as long as we extend the distribution from $[0, 1]$ to the whole $\mathbb{R}$:

$$\mathbb{E}_{p(t|\mathbf{z})}[t] = \frac{\int_{-\infty}^{\infty} \frac{1}{t^{d-1}} \exp\left(-\frac{dt_0^2}{2t^2}\right)}{\int_{-\infty}^{\infty} \frac{1}{t^d} \exp\left(-\frac{dt_0^2}{2t^2}\right)} = \frac{\frac{1}{2}\left(\frac{2}{dt_0^2}\right)^{\frac{d-2}{2}} \Gamma\left(\frac{d-2}{2}\right)}{\frac{1}{2}\left(\frac{2}{dt_0^2}\right)^{\frac{d-1}{2}} \Gamma\left(\frac{d-1}{2}\right)} = \sqrt{\frac{d}{2}} t_0 \frac{\Gamma\left(\frac{d-2}{2}\right)}{\Gamma\left(\frac{d-1}{2}\right)}$$

and similarly

$$\mathbb{E}_{p(t|\mathbf{z})}[t^2] = \frac{dt_0^2}{2} \frac{\Gamma\left(\frac{d-3}{2}\right)}{\Gamma\left(\frac{d-1}{2}\right)} = dt_0^2 \cdot \frac{1}{d-3}$$

which yields

$$\text{Var}_{p(t|\mathbf{z})}[t] = \frac{dt_0^2}{2} \cdot \left(\frac{2}{d-3} - \left(\frac{\Gamma(\frac{d-2}{2})}{\Gamma(\frac{d-1}{2})}\right)^2\right). \tag{32}$$

Again, here we extend the distribution of $p(t|\mathbf{z})$ from $[0, 1]$ to $\mathbb{R}$, as this relaxation will only increase the variance.

Finally, we use the Stirling's expansion for the gamma function to get

$$\frac{\Gamma\left(\frac{d-2}{2}\right)}{\Gamma\left(\frac{d-1}{2}\right)} = \frac{e^{-\frac{d-2}{2}} \left(\frac{d-2}{2}\right)^{\frac{d-3}{2}} \left(1 + \frac{1}{12} \cdot \frac{2}{d} + o(\frac{1}{d})\right)}{e^{-\frac{d-1}{2}} \left(\frac{d-1}{2}\right)^{\frac{d-2}{2}} \left(1 + \frac{1}{12} \cdot \frac{2}{d} + o(\frac{1}{d})\right)} = \sqrt{\frac{2}{d-1}} \left(1 + \frac{3}{4(d-1)} + o\left(\frac{1}{d}\right)\right). \tag{33}$$

Plugging Eq. (33) into Eq. (32), we derive our final estimation:

$$\text{Var}_{p(t|\mathbf{z})}[t] = \frac{dt_0^2}{2} \cdot \left(\frac{2}{d-3} - \frac{2}{d-1}\left(1 + \frac{3}{2(d-1)} + o\left(\frac{1}{d}\right)\right)\right)$$
$$= \frac{t_0^2}{d}\left(\frac{1}{2} + o\left(\frac{1}{d}\right)\right),$$

which aligns with our Statement 1.

### C.3. Approximation of $E(\mathbf{z})$

**Statement 1** (**Error of effective regression targets**). *Consider a single datapoint $\mathbf{x} \in [-1, 1]^d$, $\epsilon \sim \mathcal{N}(\mathbf{0}, \mathbf{I})$, $t \sim \mathcal{U}[0, 1]$, and $\mathbf{z} = (1 - t)\mathbf{x} + t\epsilon$ (as in Flow Matching). Define $R(\mathbf{z})$ and $R(\mathbf{z}|t)$ with the Flow Matching configuration in Table 8. Given a noisy image $\mathbf{z} = (1 - t_*)\mathbf{x} + t_*\epsilon$ produced by a given $t_*$, the mean squared error $E(\mathbf{z})$ in Eq. (10) can be approximated by*

$$E(\mathbf{z}) \approx \frac{1}{2}(1 + \sigma_{\mathrm{d}}^2) \tag{34}$$

*under the situation that the data dimension $d$ satisfies $\frac{1}{d} \ll t_*$ and $\frac{1}{d} \ll 1 - t_*$. Here, $\sigma_{\mathrm{d}}^2$ denotes the mean of squared pixel values of the dataset.*

*Derivation.* We start by the definition of $E(\mathbf{z})$:

$$E(\mathbf{z}) := \mathbb{E}_{t \sim p(t|\mathbf{z})} \|R(\mathbf{z}|t) - R(\mathbf{z})\|^2$$
$$= \mathbb{E}_{t \sim p(t|\mathbf{z})} \left\|R(\mathbf{z}|t) - \mathbb{E}_{t' \sim p(t'|\mathbf{z})}[R(\mathbf{z}|t')]\right\|^2 \tag{35}$$

Next, we compute $R(\mathbf{z}, t)$ using its definition under the Flow Matching configuration:

$$R(\mathbf{z}|t) := \mathbb{E}_{(\mathbf{x}, \epsilon) \sim p(\mathbf{x}, \epsilon|\mathbf{z}, t)}[r(\mathbf{x}, \epsilon, t)] = \mathbb{E}_{(\mathbf{x}, \epsilon) \sim p(\mathbf{x}, \epsilon|\mathbf{z}, t)}[\epsilon - \mathbf{x}] = \frac{\mathbf{z} - \mathbf{x}}{t}. \tag{36}$$

Using Eq. (36), we obtain

$$E(\mathbf{z}) = \|\mathbf{z} - \mathbf{x}\|^2 \cdot \mathrm{Var}_{t \sim p(t|\mathbf{z})}\left[\frac{1}{t}\right]. \tag{37}$$

We now compute the two terms separately. For the first term, we can rewrite it as

$$\begin{aligned}
\|\mathbf{z} - \mathbf{x}\|^2 &= t_*^2 \|\mathbf{x} - \boldsymbol{\epsilon}_*\|^2 \\
&\approx t_*^2 \left(\|\mathbf{x}\|^2 + \|\boldsymbol{\epsilon}_*\|^2\right) \\
&\approx t_*^2 \left(d\sigma_\mathrm{d}^2 + d\right) = t_*^2 d(1 + \sigma_\mathrm{d}^2).
\end{aligned} \tag{38}$$

Here, we employ the fact that $\mathbf{x} \cdot \boldsymbol{\epsilon}_* \ll \|\mathbf{x}\|\|\boldsymbol{\epsilon}_*\|$, and that $\|\boldsymbol{\epsilon}_*\| \approx \sqrt{d}$ with high probability. Also, $\sigma_\mathrm{d}^2 = \|\mathbf{x}\|^2/d$, since we assume that the dataset contains only a single data point.

For the second term, note that the variance of $p(t|\mathbf{z})$, given in Statement 1, is significantly smaller than the concentrated mean $t_*$ of $p(t|\mathbf{z})$. Thus, we approximate the variance using a first-order expansion:

$$\mathrm{Var}_{t \sim p(t|\mathbf{z})}\left[\frac{1}{t}\right] \approx \mathrm{Var}_{t \sim p(t|\mathbf{z})}\left[\frac{1}{t_*} - \frac{(t - t_*)}{t_*^2}\right] = \frac{1}{t_*^4}\mathrm{Var}_{t \sim p(t|\mathbf{z})}[t] \approx \frac{1}{t_*^2 d}. \tag{39}$$

Combining Eqs. (38) and (39), we get the estimation in Eq. (34). □

## C.4. Bound of Accumulated Error

**Statement 2 (Bound of accumulated error).** *Starting from the same noise $\mathbf{x}_0 = \mathbf{x}_0'$, consider a sampling process (Eq. (4)) of $N$ steps, with noise conditioning:*

$$\mathbf{x}_{i+1} = \kappa_i \mathbf{x}_i + \eta_i R(\mathbf{x}_i|t_i) + \zeta_i \tilde{\boldsymbol{\epsilon}}_i$$

*and without noise conditioning:*

$$\mathbf{x}_{i+1}' = \kappa_i \mathbf{x}_i' + \eta_i R(\mathbf{x}_i') + \zeta_i \tilde{\boldsymbol{\epsilon}}_i.$$

*If $\|R(\mathbf{x}_i'|t_i) - R(\mathbf{x}_i|t_i)\| / \|\mathbf{x}_i' - \mathbf{x}_i\| \leqslant L_i$ and $\|R(\mathbf{x}_i') - R(\mathbf{x}_i'|t_i)\| \leqslant \delta_i$, it can be shown that the error between the sampler outputs $\mathbf{x}_N$ and $\mathbf{x}_N'$ is bounded:*

$$\|\mathbf{x}_N - \mathbf{x}_N'\| \leqslant A_0 B_0 + A_1 B_1 + \ldots + A_{N-1} B_{N-1}, \tag{40}$$

*where*

$$A_i = \prod_{j=i+1}^{N-1} (\kappa_i + |\eta_i|L_i), \quad B_i = |\eta_i|\delta_i.$$

*Proof.* Define $a_i := \kappa_i + |\eta_i|L_i$ and $b_i := |\eta_i|\delta_i$. Then, we have:

$$\|\mathbf{x}_{i+1}' - \mathbf{x}_{i+1}\| = \left\|\kappa_i(\mathbf{x}_i' - \mathbf{x}_i) + \eta_i\left(R(\mathbf{x}_i') - R(\mathbf{x}_i|t_i)\right)\right\| \tag{41}$$

as we assume that the same noise $\tilde{\boldsymbol{\epsilon}}_i$ is added in the sampling process with and without noise conditioning.

Using the triangle inequality, this can be bounded as:

$$\|\mathbf{x}_{i+1}' - \mathbf{x}_{i+1}\| \leqslant \kappa_i\|\mathbf{x}_i' - \mathbf{x}_i\| + |\eta_i|\|R(\mathbf{x}_i') - R(\mathbf{x}_i'|t_i)\| + |\eta_i|\|R(\mathbf{x}_i'|t_i) - R(\mathbf{x}_i|t_i)\| \leqslant a_i\|\mathbf{x}_i' - \mathbf{x}_i\| + b_i. \tag{42}$$

We now use induction on $n$ to establish the bound:

$$\|\mathbf{x}_n' - \mathbf{x}_n\| \leqslant \sum_{j=0}^{n-1} \left(\prod_{k=j+1}^{n-1} a_k\right) b_j, \tag{43}$$

where $\prod_{k=j+1}^{N-1} a_k$ is defined as 1 for $j = N - 1$.

For the base case $n = 1$, we need to show:

$$\|\mathbf{x}_1' - \mathbf{x}_1\| \leqslant b_0, \tag{44}$$

which follows directly from Eq. (42) with $i = 0$.

Now, assume the bound holds for some $n$, *i.e.*

$$\|\mathbf{x}_n' - \mathbf{x}_n\| \leqslant \sum_{j=0}^{n-1} \left( \prod_{k=j+1}^{n-1} a_k \right) b_j + \left( \prod_{k=0}^{n-1} a_k \right) \|\mathbf{x}_0' - \mathbf{x}_0\|. \tag{45}$$

We prove it holds for $n + 1$. Applying Eq. (42), we obtain:

$$\|\mathbf{x}_{n+1}' - \mathbf{x}_{n+1}\| \leqslant a_n \|\mathbf{x}_n' - \mathbf{x}_n\| + b_n. \tag{46}$$

Substitute the inductive hypothesis for $\|\mathbf{x}_n' - \mathbf{x}_n\|$:

$$\|\mathbf{x}_{n+1}' - \mathbf{x}_{n+1}\| \leqslant a_n \sum_{j=0}^{n-1} \left( \prod_{k=j+1}^{n-1} a_k \right) b_j + b_n = \sum_{j=0}^{n} \left( \prod_{k=j+1}^{n} a_k \right) b_j. \tag{47}$$

Thus, the bound holds for $n + 1$. By induction, the bound holds for all $n$. Taking $n = N$ yields the desired result. $\square$

## D. Derivation of Coefficients for Different Denoising Generative Models

In this section, we build upon our formulation in Section 3.1 to express common diffusion models—iDDPM (Nichol & Dhariwal, 2021), DDIM (Song et al., 2021a), EDM (Karras et al., 2022), Flow Matching (FM) (Lipman et al., 2023; Liu et al., 2023), and our uEDM Model—using a unified notation. The coefficients corresponding to each model are summarized in Table 8 and Table 9, followed by a concise derivation of their formulations.

### D.1. iDDPM

The loss function of iDDPM (Nichol & Dhariwal, 2021) in DDPM (Ho et al., 2020)'s notation is

$$\mathcal{L}_{\text{simple}} = \mathbb{E}_{t,\mathbf{x}_0,\boldsymbol{\epsilon}} \left[ \left\| \boldsymbol{\epsilon} - \boldsymbol{\epsilon}_{\boldsymbol{\theta}}(\sqrt{\bar{\alpha}_t}\mathbf{x}_0 + \sqrt{1 - \bar{\alpha}_t}\boldsymbol{\epsilon}, t) \right\|^2 \right].$$

This can be directly translated into our notation:

$$\mathcal{L}(\boldsymbol{\theta}) = \mathbb{E}_{\mathbf{x},\boldsymbol{\epsilon},t} \left[ w(t) \| \text{NN}_{\boldsymbol{\theta}}(\mathbf{z}|c_{\text{noise}}(t)) - r(\mathbf{x}, \boldsymbol{\epsilon}, t) \|^2 \right],$$

where we have the coefficients

$$a(t) = \sqrt{\bar{\alpha}(t)}, b(t) = \sqrt{1 - \bar{\alpha}(t)}, c(t) = 0, d(t) = 1 \tag{48}$$

and with the training weighting and distribution of $t$ being

$$w(t) = 1, \quad \text{and} \quad p(t) = \mathcal{U}\{1, \ldots, T\}. \tag{49}$$

Notice the presence of the diffusion schedule $\bar{\alpha}(t)$ inside the coefficients. We adapt a modified version of the cosine schedule in Nichol & Dhariwal (2021):

$$\bar{\alpha}(t) = \frac{1}{2} \left( 1 + \cos \frac{\pi t}{T} \right), \tag{50}$$

Table 8: The coefficients of different models. For iDDPM, we assume a cosine diffusion schedule $\bar{\alpha}(t)$. For both iDDPM and DDIM we follow the original notation of DDPM (Ho et al., 2020). Also note that for EDM, all coefficients are calculated according to first-order ODE solver, and in the final step we need to multiply the output by $\sigma_\mathrm{d}$ to get the final image. See Appendix D for more details and derivations.

| | iDDPM | DDIM | EDM | FM |
|---|---|---|---|---|
| **Training** | | | | |
| $a(t)$ | $\sqrt{\bar{\alpha}(t)}$ | $\sqrt{\bar{\alpha}(t)}$ | $\frac{1}{\sqrt{t^2+\sigma_\mathrm{d}^2}}$ | $1-t$ |
| $b(t)$ | $\sqrt{1-\bar{\alpha}(t)}$ | $\sqrt{1-\bar{\alpha}(t)}$ | $\frac{t}{\sqrt{t^2+\sigma_\mathrm{d}^2}}$ | $t$ |
| $c(t)$ | $0$ | $0$ | $\frac{t}{\sigma_\mathrm{d}\sqrt{t^2+\sigma_\mathrm{d}^2}}$ | $-1$ |
| $d(t)$ | $1$ | $1$ | $-\frac{\sigma_\mathrm{d}}{\sqrt{t^2+\sigma_\mathrm{d}^2}}$ | $1$ |
| $w(t)$ | $1$ | $1$ | $1$ | $1$ |
| $p_t$ | $\mathcal{U}\{1,\dots,T\}$ | $\mathcal{U}\{1,\dots,T\}$ | $\exp\mathcal{N}(-1.2, 1.2^2)$ [7] | $\mathcal{U}[0,1]$ |
| **Sampling** | | | | |
| $\kappa_i$ | $\sqrt{\frac{\bar{\alpha}_{i+1}}{\bar{\alpha}_i}}$ | $\sqrt{\frac{\bar{\alpha}_{i+1}}{\bar{\alpha}_i}}$ | $\sqrt{\frac{\sigma_\mathrm{d}^2+t_i^2}{\sigma_\mathrm{d}^2+t_{i+1}^2}}\left(1-\frac{t_i(t_i-t_{i+1})}{t_i^2+\sigma_\mathrm{d}^2}\right)$ | $0$ |
| $\eta_i$ | $\frac{1}{\sqrt{1-\bar{\alpha}_i}}\left(\sqrt{\frac{\bar{\alpha}_i}{\bar{\alpha}_{i+1}}}-\sqrt{\frac{\bar{\alpha}_{i+1}}{\bar{\alpha}_i}}\right)$ | $\sqrt{1-\bar{\alpha}_{i+1}}-\sqrt{\frac{\bar{\alpha}_{i+1}}{\bar{\alpha}_i}(1-\bar{\alpha}_i)}$ | $\frac{\sigma_\mathrm{d}(t_i-t_{i+1})}{\sqrt{(t_i^2+\sigma_\mathrm{d}^2)(t_{i+1}^2+\sigma_\mathrm{d}^2)}}$ | $t_{i+1}-t_i$ |
| $\zeta_i$ | $\sqrt{\left(1-\frac{\bar{\alpha}_i}{\bar{\alpha}_{i+1}}\right)\frac{1-\bar{\alpha}_{i+1}}{1-\bar{\alpha}_i}}$ | $0$ | $0$ | $0$ |
| Schedule $t_{0\sim N}$ | $t_i=\frac{N-i}{N}\cdot T$ | $t_i=\frac{N-i}{N}\cdot T$ | $t_i=\left(t_\mathrm{max}^{\frac{1}{\rho}}+\frac{i}{N}\left(t_\mathrm{min}^{\frac{1}{\rho}}-t_\mathrm{max}^{\frac{1}{\rho}}\right)\right)^\rho$ | $t_i=1-\frac{i}{N}$ |
| **Parameters** | | | | |
| | $\bar{\alpha}(t)=\frac{1}{2}\left(1+\cos\frac{\pi t}{T}\right)$ | $\bar{\alpha}(t)=\prod_{i=0}^{t-1}\left(1-k_1-k_2\frac{i}{T-1}\right)$ | $\sigma_\mathrm{d}=0.5, \rho=7$ | |
| | $\bar{\alpha}_i:=\bar{\alpha}(t_i)$ | $\bar{\alpha}_i:=\bar{\alpha}(t_i)$ | $t_\mathrm{max}=80, t_\mathrm{min}=0.002$ | |
| | $T=4000$ | $T=1000$ | | |
| | | $k_1=10^{-4}, k_2=2\times10^{-2}$ | | |

where $T=4000$ is the total number of diffusion steps during training.

Next, consider the sampling process, which in their notations is iteratively given by

$$\mathbf{x}_{t-1}=\frac{1}{\sqrt{\alpha_t}}\left(\mathbf{x}_t-\frac{1-\alpha_t}{\sqrt{1-\bar{\alpha}_t}}\boldsymbol{\epsilon}_{\boldsymbol{\theta}}(\mathbf{x}_t,t)\right)+\sqrt{\frac{1-\bar{\alpha}_{t-1}}{1-\bar{\alpha}_t}}\beta_t\mathbf{z},$$

and $\mathbf{z}\sim\mathcal{N}(\mathbf{0},\boldsymbol{I})$ is a standard Gaussian random noise. It is also straightforward to translate this sampling equation into our notation:

$$\kappa_i=\sqrt{\frac{\bar{\alpha}_{i+1}}{\bar{\alpha}_i}}, \eta_i=\frac{1}{\sqrt{1-\bar{\alpha}_i}}\left(\sqrt{\frac{\bar{\alpha}_i}{\bar{\alpha}_{i+1}}}-\sqrt{\frac{\bar{\alpha}_{i+1}}{\bar{\alpha}_i}}\right), \zeta_i=\sqrt{\left(1-\frac{\bar{\alpha}_i}{\bar{\alpha}_{i+1}}\right)\frac{1-\bar{\alpha}_{i+1}}{1-\bar{\alpha}_i}}, \tag{51}$$

and

$$t_i=\frac{N-i}{N}\cdot T. \tag{52}$$

This will give the first column in Table 8.

## D.2. DDIM

DDIM (Song et al., 2021a) shares the training process with DDPM (Ho et al., 2020). However, we choose to use the linear schedule for $\bar{\alpha}(t)$, to demonstrate the generality of our scheme. This schedule has the form

$$\bar{\alpha}(t) = \prod_{i=0}^{t-1}\left(1 - k_1 - k_2\frac{i}{T-1}\right),\tag{53}$$

where $k_1 = 10^{-4}$ and $k_2 = 2 \times 10^{-2}$, and $T = 1000$ is the total number of diffusion steps during training.

The sampling process is given by

$$\mathbf{x}_{t-1} = \sqrt{\bar{\alpha}_{t-1}}\left(\frac{x_t - \sqrt{1-\bar{\alpha}_t}\boldsymbol{\epsilon_\theta}(\mathbf{x}_t, t)}{\sqrt{\bar{\alpha}_t}}\right) + \sqrt{1 - \bar{\alpha}_{t-1}}\boldsymbol{\epsilon_\theta}(\mathbf{x}_t, t)$$

which is obtained by substituting $\sigma_t = 0$ in their notation. This is again straightforward to translate into our notation:

$$\kappa_i = \sqrt{\frac{\bar{\alpha}_{i+1}}{\bar{\alpha}_i}}, \eta_i = \sqrt{1 - \bar{\alpha}_{i+1}} - \sqrt{\frac{\bar{\alpha}_{i+1}}{\bar{\alpha}_i}(1 - \bar{\alpha}_i)}, \zeta_i = 0,\tag{54}$$

and

$$t_i = \frac{N-i}{N}\cdot T.\tag{55}$$

These give the second column in Table 8.

Moreover, we can consider the generalized sampler proposed by Song et al. (2021a), which contains an adjustable parameter $\lambda \in [0, 1]$. In their original notation, the sampler can be written as

$$\mathbf{x}_{t-1} = \sqrt{\bar{\alpha}_{t-1}}\left(\frac{x_t - \sqrt{1-\bar{\alpha}_t}\boldsymbol{\epsilon_\theta}(\mathbf{x}_t, t)}{\sqrt{\bar{\alpha}_t}}\right) + \sqrt{1 - \bar{\alpha}_{t-1} - \lambda^2\sigma_t^2}\boldsymbol{\epsilon_\theta}(\mathbf{x}_t, t) + \lambda\sigma_t\boldsymbol{\epsilon}_t,$$

where

$$\sigma_t := \sqrt{\frac{(\bar{\alpha}_{t-1} - \bar{\alpha}_t)(1 - \bar{\alpha}_{t-1})}{\bar{\alpha}_{t-1}(1 - \bar{\alpha}_t)}}$$

and $\boldsymbol{\epsilon}_t$ is an independent Gaussian random noise. In our formulation, it can be equivalently written as

$$\begin{cases} \kappa_i = \sqrt{\dfrac{\bar{\alpha}_{i+1}}{\bar{\alpha}_i}} \\[2mm] \eta_i = \sqrt{1 - \bar{\alpha}_{i+1} - \lambda^2\dfrac{(\bar{\alpha}_{i+1} - \bar{\alpha}_i)(1 - \bar{\alpha}_{i+1})}{\bar{\alpha}_{i+1}(1 - \bar{\alpha}_i)}} - \sqrt{\dfrac{\bar{\alpha}_{i+1}}{\bar{\alpha}_i}(1 - \bar{\alpha}_i)} \\[2mm] \zeta_i = \lambda\sqrt{\dfrac{(\bar{\alpha}_{i+1} - \bar{\alpha}_i)(1 - \bar{\alpha}_{i+1})}{\bar{\alpha}_{i+1}(1 - \bar{\alpha}_i)}} \end{cases}.\tag{56}$$

These expressions are used in our experiment of the "interpolate sampler" in Section 6.2. One can verify that when $\lambda = 1$, the coefficients $\kappa_i, \eta_i$ and $\zeta_i$ will be the same as iDDPM (Eq. (51)); and when $\lambda = 0$, the coefficients will be the same as DDIM (Eq. (54)).

---

[7]Here, we use the notation $\exp\mathcal{N}(\mu, \sigma^2)$ to denote the distribution of $\exp(u)$, where $u \sim \mathcal{N}(\mu, \sigma^2)$.

## D.3. EDM

The original EDM (Karras et al., 2022) training objective is given by

$$\mathcal{L}(\boldsymbol{\theta}) = \mathbb{E}_{\mathbf{x}, \boldsymbol{\epsilon}, t}\Big[\lambda(t)\|\boldsymbol{D}_{\boldsymbol{\theta}}(\mathbf{x} + t\boldsymbol{\epsilon}|t) - \mathbf{x}\|^2\Big], \tag{57}$$

where $\boldsymbol{D}_{\boldsymbol{\theta}}$ is formed by the *raw network* $\mathrm{NN}_{\boldsymbol{\theta}}$ wrapped with a precondition:

$$\boldsymbol{D}_{\boldsymbol{\theta}}(\mathbf{z}_{\boldsymbol{D}}|t) = c_{\mathrm{skip}}(t)\mathbf{z}_{\boldsymbol{D}} + c_{\mathrm{out}}(t)\mathrm{NN}_{\boldsymbol{\theta}}(c_{\mathrm{in}}(t)\mathbf{z}_{\boldsymbol{D}}|t)$$

where $\mathbf{z}_{\boldsymbol{D}} = \mathbf{x} + t\boldsymbol{\epsilon}$. Here we directly use $t$ instead of $c_{\mathrm{noise}}(t)$ in the original notation.

As mentioned in Section 3.1, we will consider the regression target with respect to $\mathrm{NN}_{\boldsymbol{\theta}}$ instead of $\boldsymbol{D}_{\boldsymbol{\theta}}$ and absorb the coefficients $c_{\mathrm{skip}}(t)$, $c_{\mathrm{in}}(t)$ and $c_{\mathrm{out}}(t)$ into the training process. To achieve that, we define $\mathbf{z} := c_{\mathrm{in}}(t)(\mathbf{x} + t\boldsymbol{\epsilon})$. Then, we can get an equivalent training objective for $\mathrm{NN}_{\boldsymbol{\theta}}$ given by

$$\mathcal{L}(\boldsymbol{\theta}) = \mathbb{E}_{\mathbf{x}, \boldsymbol{\epsilon}, t}\Big[w(t)\|\mathrm{NN}_{\boldsymbol{\theta}}(\mathbf{z}|t) - r(\mathbf{x}, \boldsymbol{\epsilon}, t)\|^2\Big],$$

where

$$\begin{cases} z = c_{\mathrm{in}}(t)\mathbf{x} + t c_{\mathrm{in}}(t)\boldsymbol{\epsilon} \\ w(t) = \lambda(t)c_{\mathrm{out}}(t)^2 \\ r(\mathbf{x}, \boldsymbol{\epsilon}, t) = \frac{1}{c_{\mathrm{out}}(t)} \cdot (\mathbf{x} - c_{\mathrm{skip}}(t)(\mathbf{x} + t\boldsymbol{\epsilon})) = \frac{1 - c_{\mathrm{skip}}(t)}{c_{\mathrm{out}}(t)}\mathbf{x} - \frac{t c_{\mathrm{skip}}(t)}{c_{\mathrm{out}}(t)}\boldsymbol{\epsilon} \end{cases} \tag{58}$$

Now, we can plug in the specific expressions

$$c_{\mathrm{in}}(t) = \frac{1}{\sqrt{\sigma_{\mathrm{d}}^2 + t^2}}, c_{\mathrm{out}}(t) = \frac{\sigma_{\mathrm{d}} t}{\sqrt{\sigma_{\mathrm{d}}^2 + t^2}}, c_{\mathrm{skip}}(t) = \frac{\sigma_{\mathrm{d}}^2}{\sigma_{\mathrm{d}}^2 + t^2}, \lambda(t) = \frac{\sigma_{\mathrm{d}}^2 + t^2}{\sigma_{\mathrm{d}}^2 t^2}$$

and $\sigma_{\mathrm{d}} = 0.5$ to get the coefficients

$$a(t) = \frac{1}{\sqrt{\sigma_{\mathrm{d}}^2 + t^2}}, b(t) = \frac{t}{\sqrt{\sigma_{\mathrm{d}}^2 + t^2}}, c(t) = \frac{t}{\sigma_{\mathrm{d}}\sqrt{\sigma_{\mathrm{d}}^2 + t^2}}, d(t) = -\frac{\sigma_{\mathrm{d}}}{\sqrt{\sigma_{\mathrm{d}}^2 + t^2}}, \tag{59}$$

and $w(t) = 1$. Also note that $p(t)$ is given explicitly by the log-norm schedule $\exp\mathcal{N}(-1.2, 1.2^2)$. This completes the discussion of the training process.

The (first-order) sampling process is given by

$$\mathbf{x}_{\boldsymbol{D}, i+1} = \mathbf{x}_{\boldsymbol{D}, i} + (t_{i+1} - t_i)\frac{\mathbf{x}_{\boldsymbol{D}, i} - (c_{\mathrm{skip}}(t_i)\mathbf{x}_{\boldsymbol{D}, i} + c_{\mathrm{out}}(t_i)\mathrm{NN}_{\boldsymbol{\theta}}(c_{\mathrm{in}}(t_i)\mathbf{x}_{\boldsymbol{D}, i}|t_i))}{t_i}$$

Here we use the suffix $\boldsymbol{D}$ to denote this is the sampling process corresponding to $\boldsymbol{D}_{\boldsymbol{\theta}}$. Since we also have to remove the external conditioning in the sampling process, we should let $\mathbf{x}_i = c_{\mathrm{in}}(t_i)\mathbf{x}_{\boldsymbol{D}, i}$ and rewrite the sampling equation using $\mathbf{x}_i$:

$$\mathbf{x}_{i+1} = \frac{t_{i+1}}{t_i} \cdot \frac{c_{\mathrm{in}}(t_{i+1})}{c_{\mathrm{in}}(t_i)}\left(1 - \frac{t_{i+1} - t_i}{t_{i+1}}c_{\mathrm{skip}}(t_i)\right)\mathbf{x}_i + \frac{t_i - t_{i+1}}{t_i}c_{\mathrm{out}}(t_i)c_{\mathrm{in}}(t_{i+1})\mathrm{NN}_{\boldsymbol{\theta}}(\mathbf{x}_i|t_i)$$

This then gives the general sampling coefficients

$$\kappa_i = \frac{t_{i+1}}{t_i} \cdot \frac{c_{\mathrm{in}}(t_{i+1})}{c_{\mathrm{in}}(t_i)}\left(1 - \frac{t_{i+1} - t_i}{t_{i+1}}c_{\mathrm{skip}}(t_i)\right), \eta_i = \frac{t_i - t_{i+1}}{t_i}c_{\mathrm{out}}(t_i)c_{\mathrm{in}}(t_{i+1}), \zeta_i = 0. \tag{60}$$

Then, we can plug in the explicit expressions of $c_{\mathrm{in}}(t_i)$, $c_{\mathrm{skip}}(t_i)$ and $c_{\mathrm{out}}(t_i)$ to get the final coefficients

$$\kappa_i = \sqrt{\frac{\sigma_{\mathrm{d}}^2 + t_i^2}{\sigma_{\mathrm{d}}^2 + t_{i+1}^2}}\left(1 + \frac{t_i(t_{i+1} - t_i)}{t_i^2 + \sigma_{\mathrm{d}}^2}\right), \eta_i = -\frac{\sigma_{\mathrm{d}}(t_{i+1} - t_i)}{\sqrt{(t_{i+1}^2 + \sigma_{\mathrm{d}}^2)(t_i^2 + \sigma_{\mathrm{d}}^2)}}, \zeta_i = 0. \tag{61}$$

Moreover, notice that due to our change-of-variable during the removal of external conditioning, $\mathbf{x}_N$ is defined as $c_{\text{in}}(t_N)\mathbf{x}_{D,N}$. But the sampling algorithm ensures $\mathbf{x}_{D,N}$ to match the data distribution, instead of $\mathbf{x}_N$. Thus, we have to multiply the output by $\sigma_{\text{d}}$ to get the final image, as mentioned in the caption of Table 8.

Finally, the sampling time step is also explicitly given in Karras et al. (2022), so we can directly use it here:

$$t_i = \begin{cases} \left( \frac{80^{\frac{1}{7}} \cdot (N-i-1) + 0.002^{\frac{1}{7}} \cdot i}{N-1} \right)^7 & \text{if } i < N \\ 0 & \text{if } i = N \end{cases}. \tag{62}$$

These together give the coefficients for EDM, which are shown in the third column of Table 8.

### D.4. Flow Matching

The training process of FM (Lipman et al., 2023) is given by

$$\mathcal{L}(\boldsymbol{\theta}) = \mathbb{E}_{\mathbf{x},\boldsymbol{\epsilon},t}\left[ \left\| \mathbf{v}_{\boldsymbol{\theta}}(t\boldsymbol{\epsilon} + (1-t)\mathbf{x}, t) - (\boldsymbol{\epsilon} - \mathbf{x}) \right\|^2 \right].$$

This can be directly translated into our notation:

$$a(t) = 1-t, b(t) = t, c(t) = -1, d(t) = 1, \tag{63}$$

and with $w(t) = 1$ and $p(t) = \mathcal{U}([0,1])$. The sampling process is given by solving the ODE

$$\frac{d\mathbf{x}}{dt} = \mathbf{v}_{\boldsymbol{\theta}}(\mathbf{x}, t)$$

from $t = 1$ to $t = 0$. Since we assume using a first-order method (*i.e.* Euler method), the sampling equation is

$$\mathbf{x}_{i+1} = \mathbf{x}_i + \mathbf{v}_{\boldsymbol{\theta}}(\mathbf{x}_i, t_i) \cdot (t_{i+1} - t_i).$$

This will give

$$\kappa_i = 0, \eta_i = t_{i+1} - t_i, \zeta_i = 0 \tag{64}$$

as well as the sampling time

$$t_i = \frac{N-i}{N}, \tag{65}$$

as in the fourth column of Table 8.

### D.5. Our uEDM Model in the Formulation

Introduced in Section 5, the uEDM model designed by us is a modified version of EDM (Karras et al., 2022). The only modification is that we change $c_{\text{in}}(t)$ and $c_{\text{out}}(t)$ by

$$\begin{cases} c_{\text{in}}(t) = \dfrac{1}{\sqrt{t^2 + \sigma_{\text{d}}^2}} \\ c_{\text{out}}(t) = \dfrac{t\sigma_{\text{d}}}{\sqrt{t^2 + \sigma_{\text{d}}^2}} \end{cases} \longrightarrow \begin{cases} c_{\text{in}}(t) = \dfrac{1}{\sqrt{t^2 + 1}} \\ c_{\text{out}}(t) = 1 \end{cases}$$

and remain all other configurations the same as the original EDM model.

In Appendix D.3, we have already derived the general form of the coefficients of EDM with functions $c_{\text{in}}(t)$, $c_{\text{out}}(t)$, $c_{\text{skip}}(t)$ and $\lambda(t)$ in Eqs. (58) and (60). Plugging in the new set of these functions, we can then derive the coefficients of uEDM, as shown in Table 9.

Table 9: Comparison of coefficients of EDM and our uEDM.

| Coefficients | $a(t)$ | $b(t)$ | $c(t)$ | $d(t)$ | $w(t)$ | $\kappa_i$ | $\eta_i$ |
|---|---|---|---|---|---|---|---|
| EDM | $\dfrac{1}{\sqrt{t^2 + \sigma_{\mathrm{d}}^2}}$ | $\dfrac{t}{\sqrt{t^2 + \sigma_{\mathrm{d}}^2}}$ | $\dfrac{t}{\sigma_{\mathrm{d}}\sqrt{t^2 + \sigma_{\mathrm{d}}^2}}$ | $-\dfrac{\sigma_{\mathrm{d}}}{\sqrt{t^2 + \sigma_{\mathrm{d}}^2}}$ | $1$ | $\sqrt{\dfrac{\sigma_{\mathrm{d}}^2 + t_i^2}{\sigma_{\mathrm{d}}^2 + t_{i+1}^2}}\left(1 - \dfrac{t_i(t_i - t_{i+1})}{t_i^2 + \sigma_{\mathrm{d}}^2}\right)$ | $\dfrac{\sigma_{\mathrm{d}}(t_i - t_{i+1})}{\sqrt{(t_i^2 + \sigma_{\mathrm{d}}^2)(t_{i+1}^2 + \sigma_{\mathrm{d}}^2)}}$ |
| uEDM | $\dfrac{1}{\sqrt{t^2 + 1}}$ | $\dfrac{t}{\sqrt{t^2 + 1}}$ | $\dfrac{t^2}{t^2 + \sigma_{\mathrm{d}}^2}$ | $-\dfrac{t\sigma_{\mathrm{d}}^2}{t^2 + \sigma_{\mathrm{d}}^2}$ | $\dfrac{\sigma_{\mathrm{d}}^2 + t^2}{\sigma_{\mathrm{d}}t}$ | $\sqrt{\dfrac{t_i^2 + 1}{t_{i+1}^2 + 1}}\left(1 - \dfrac{t_i(t_i - t_{i+1})}{t_i^2 + \sigma_{\mathrm{d}}^2}\right)$ | $\dfrac{t_i - t_{i+1}}{t_i\sqrt{t_{i+1}^2 + 1}}$ |

# E. Additional Samples

Beyond the comparison shown in Fig. 5 for noise-conditional and noise-unconditional models, we also provide additional samples for other models, on other datasets, or in class-conditional settings. We use the same configuration as in Table 2. Figs. 10 and 11 show the samples of ICM and ECM on CIFAR-10 with both 1 and 2 inference steps. Fig. 12 show the samples of FM on ImageNet 32×32 with both Euler and EDM-Heun sampler. Fig. 13 shows the samples of FM and EDM on CIFAR-10 in a class-conditional setting.

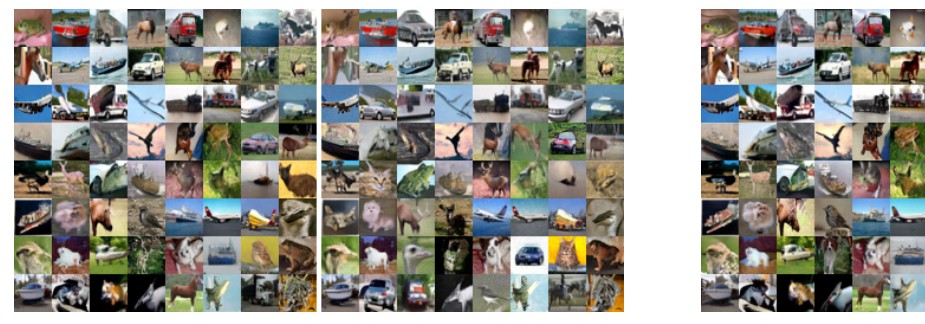

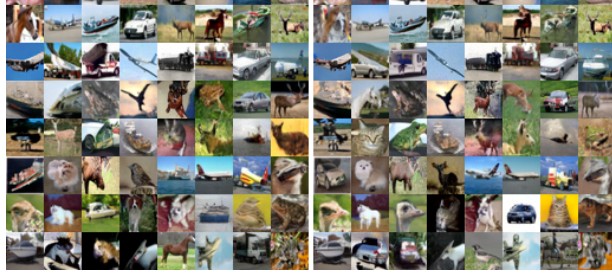

(a) ICM 1 step (FID: $3.37 \rightarrow 12.03$)        (b) ICM 2 step (FID: $2.59 \rightarrow 3.57$)

Figure 10: Samples generated by ICM on CIFAR-10. From left to right: 1 step w/ $t$, 1 step w/o $t$, 2 step w/ $t$, 2 step w/o $t$. All corresponding samples use the same noise.

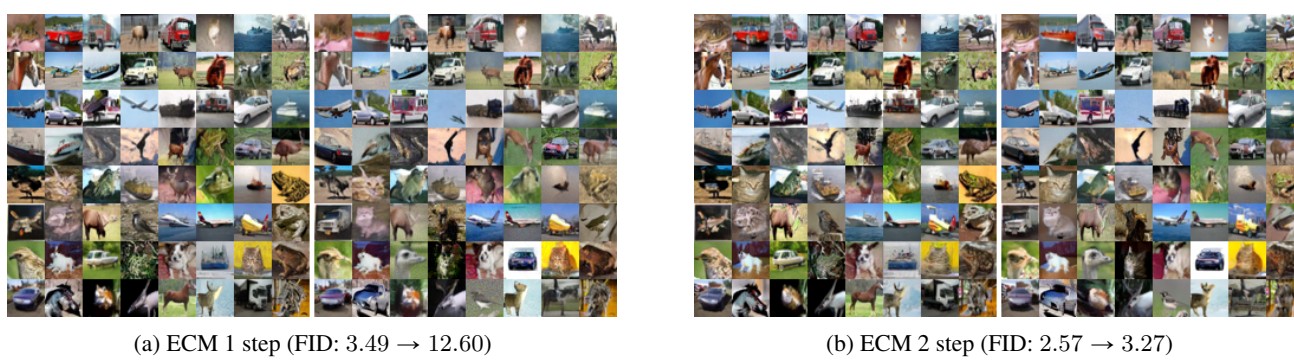

(a) ECM 1 step (FID: $3.49 \rightarrow 12.60$)        (b) ECM 2 step (FID: $2.57 \rightarrow 3.27$)

Figure 11: Samples generated by ECM on CIFAR-10. From left to right: 1 step w/ $t$, 1 step w/o $t$, 2 step w/ $t$, 2 step w/o $t$. All corresponding samples use the same noise.

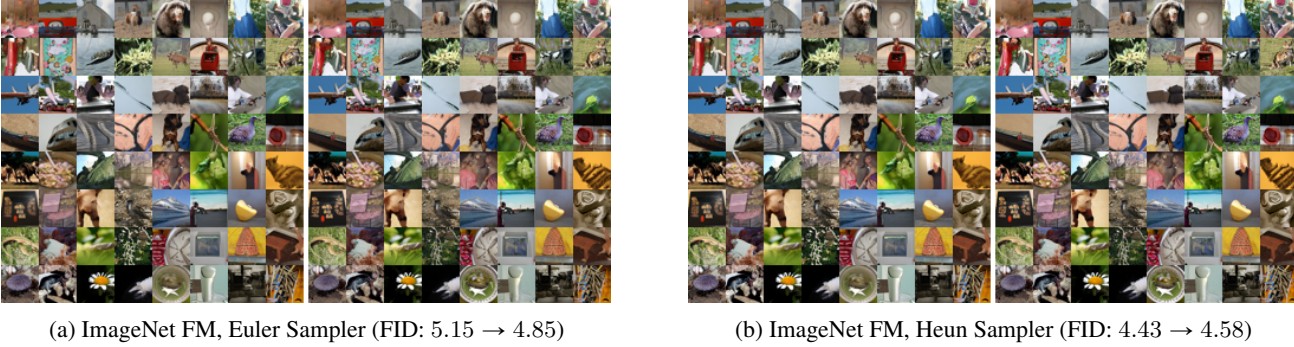

(a) ImageNet FM, Euler Sampler (FID: $5.15 \rightarrow 4.85$)      (b) ImageNet FM, Heun Sampler (FID: $4.43 \rightarrow 4.58$)

Figure 12: Samples generated by FM on ImageNet $32 \times 32$ with Euler and EDM-Heun sampler. From left to right: Euler w/ $t$, Euler w/o $t$, Heun w/ $t$, Heun w/o $t$. All corresponding samples use the same noise.

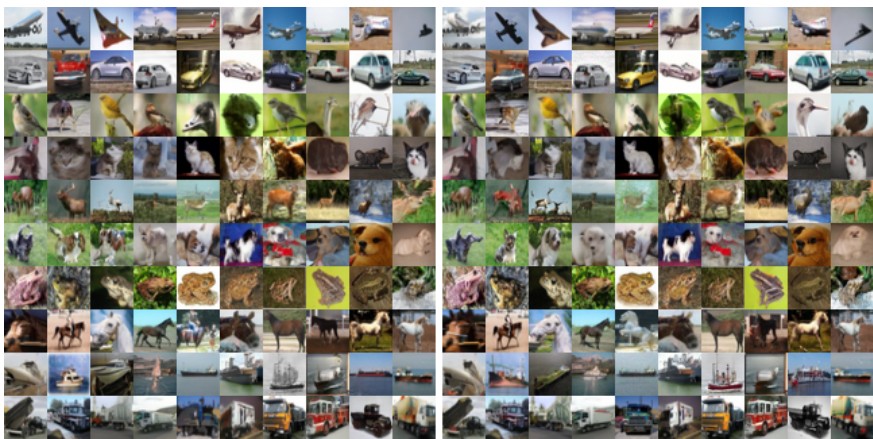

(a) Class-conditional FM (FID: 2.72 → 2.55)

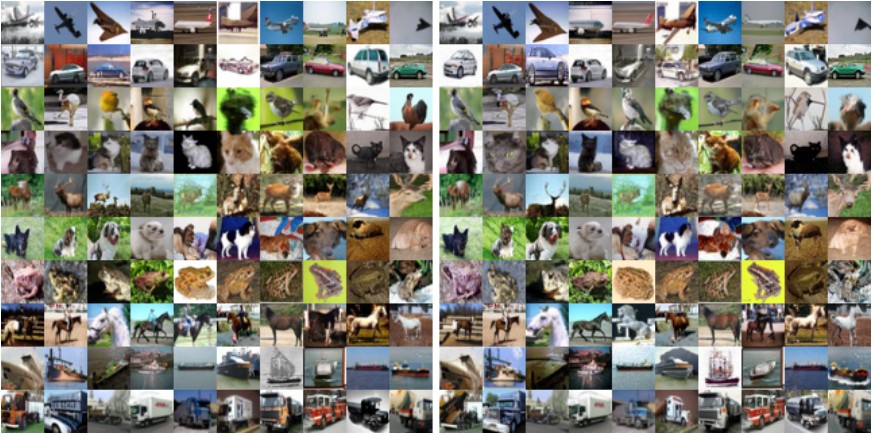

(b) Class-conditional EDM (FID: 1.76 → 3.11)

Figure 13: Class-conditional samples generated by FM and EDM on CIFAR-10. In rasterized order: FM w/ $t$, FM w/o $t$, EDM w/ $t$, EDM w/o $t$. All corresponding samples use the same noise and the same label.

