# OpenReview forum: "Is Noise Conditioning Necessary for Denoising Generative Models?"
_ICML.cc/2025/Conference — ICML 2025 poster_

### Official Review · Reviewer_FoCJ · 2025-03-09

**Overall Recommendation:** 3

**Summary:**

This paper investigates the necessity of noise conditioning in diffusion models. It provides a theoretical analysis of the effects of removing noise conditioning and presents error bounds. The analysis shows that, under mild conditions, the errors resulting from the removal of noise conditioning are relatively small. The paper further conducts experiments to validate these findings. The results indicate that while noise conditioning is beneficial for enhancing sample quality, it is not a critical factor for denoising generative models.

## update after rebuttal

I thank the authors for their responses. I have read all reviews and authors' responses. I will maintain my scores.

**Claims And Evidence:**

The claims are generally clear and supported by theoretical and empirical analysis. I only have a few questions

1. Without the noise conditioning, how are the models trained? That is, is Eq. 7 the objective for training? If yes, how do we sample $\mathbf{z}$ and compute the expectation? If no, is it that we still train models with Eq. 2, but only replace $NN(\mathbf{z}|t)$ with $NN(\mathbf{z})$?
2. If we still use Eq.2 for training and only replace $NN(\mathbf{z}|t)$ with $NN(\mathbf{z})$, we actually still use the noise conditioning. That is, the t is implicitly used since $\mathbf{z} = a(t)\mathbf{x} + b(t)\epsilon$.

**Essential References Not Discussed:**

Related works are sufficiently discussed.

**Experimental Designs Or Analyses:**

The experiment results are sound. One question I have is

1. The paper states that the error bounds are related to the feature dimension $d$. In experiments of image sets, the $d$ is much greater than $1/t$, so the statements hold. It would also be helpful to vary $d$ and see how the error bounds are impacted.

**Methods And Evaluation Criteria:**

The proposed methods make sense.

**Other Comments Or Suggestions:**

NA

**Other Strengths And Weaknesses:**

NA

**Questions For Authors:**

1. I am curious about the objective of removing the noise conditioning. That is, it seems that it will not reduce training/inference time too much, but the model performance may be impacted, i.e., in Table 2, most methods are worse.

**Relation To Broader Scientific Literature:**

The paper mainly explores the impact of noise conditioning in denoising diffusion generative models. Its analysis and results can be generalized to many different variants of diffusion models, e.g., flow matching and consistency models.

**Theoretical Claims:**

I checked the proof in Appendix B, and did not find issues.

---

> ### Author Rebuttal · Authors · 2025-04-01
>
> Thanks a lot for the insightful feedback and the supportive comments to our work!
>
> **1.Definition & Benefits of removing noise conditioning**
>
> **Reviewer FoCJ:** `Without the noise conditioning, how are the models trained? … is it that we still train models with Eq. 2, but only replace `$NN_{\theta}(\mathbf{z}\mid t)$` with `$NN_{\theta}(\mathbf{z})$?
>
> Regarding the question: Correct. Without noise conditioning, the neural network is still trained with Eq.2 but only with $NN_{\theta}(\mathbf{z}|t)$ replaced with $NN_{\theta}(\mathbf{z})$.
>
> **Reviewer FoCJ:** `… we actually still use the noise conditioning. That is, the t is implicitly used since` $z=a(t)x+b(t)\epsilon$.
>
> This is the main message we wanted to convey: the noisy image contains information of the noise level $t$, and this is why *explicit* noise conditioning in the neural net is not needed.
>
> **Reviewer FoCJ:** `I am curious about the objective of removing the noise conditioning`.
>
> Our motivation actually roots from the curiosity on the necessity of noise conditioning, which is the “common wisdom” in most of the previous works on denoising generative models. In our opinion, the value of removing noise conditioning lies in challenging common sense and also lies in its theoretical meaning. Finally, as a direct result, some models (such as DiT+FM, as we share with Reviewer gvd2 and 7y3v) can enjoy improvement in removal of noise conditioning.
>
> **2. Low-Dimensional behavior**
>
> Reviewer FoCJ suggests to investigate model behavior in the senario where dimension $d$ is low, since our theory assumes a large enough $d$ value. This suggestion is very insightful.
>
> Inspired by that, we explored low-dimensional cases, where the approximation $d\gg 1$ does not hold and our theoretical analysis is ineffective. Specifically, we did experiments on the toy “two moons” dataset (which has $d=2$) with Flow Matching (i.e. Flowing from standard Gaussian to the two-moons dataset) models, and visualize the generated samples in the noise-conditional and unconditional settings. The results are shown in the figure below:
>
> [reb — Postimages](https://postimg.cc/dZwWS6hF)
>
> By visualizing the generated samples, one can see that removing noise conditioning leads to a significant performance drop in low-dimensional cases, preventing proper modeling of the distribution.

---

### Official Review · Reviewer_7y3v · 2025-03-12

**Overall Recommendation:** 2

**Summary:**

This paper investigates whether diffusion models, which are typically noise-conditioning networks, can be converted into noise-unconditional networks. It finds that many models are not significantly affected by the removal of noise conditioning, and in the case of Rectified Flow (RF) models, performance even improves. The paper provides a theoretical analysis suggesting that removing noise conditioning does not introduce significant errors. Additionally, it proposes a noise-unconditional variant of EDM that maintains FID performance.

**Claims And Evidence:**

The core idea of the paper is strong, but the analysis lacks exploration across a broader set of models. The experimental setup has low variance, raising concerns that some results might be due to chance. The model-wise analysis is limited, as the models selected are among the simplest available.

One issue is that diffusion sampling often involves classifier-free guidance, which introduces instability through extrapolation in $x_t$. Noise-unconditioning may exacerbate these instabilities. For example, Pan et al. (2023) shows that increasing classifier-free guidance from 1 to 2 to 3 significantly degrades performance in a diffusion inversion setting. Similarly, models like Latent Diffusion Models (LDM) and DiT might exhibit unpredictable behavior under noise-unconditioning.

Another key concern is why 1-RF performs better without noise conditioning. It is likely due to 1-RF's nature: it inherently learns the shortest path between the data distribution and Gaussian noise, which aligns well with the diffusion ODE trajectory, making noise conditioning unnecessary. However, this explanation is missing from the main text.

Despite these gaps, Figures 2, 3, and 4 effectively support the logical flow between the paper’s core ideas and conclusions.

Refs:
Pan, Zhihong, et al. "Effective real image editing with accelerated iterative diffusion inversion." ICCV 2023.

**Essential References Not Discussed:**

The difference between DDIM failing and 1-RF succeeding likely stems from DDIM's trajectory curvature. A more detailed analysis of this could strengthen the paper.

Here's one paper that potentially helps discussion:

Lee, Sangyun, Beomsu Kim, and Jong Chul Ye. "Minimizing trajectory curvature of ODE-based generative models." ICML 2023.

**Experimental Designs Or Analyses:**

Sections 4.2 and 4.3, along with Figure 3, focus primarily on FM models. However, to support similar claims for DDIM and EDM, equivalent experiments should be conducted for those models.
As mentioned earlier, the analysis does not sufficiently highlight model-specific differences, making some conclusions appear oversimplified.

**Methods And Evaluation Criteria:**

The evaluation is somewhat limited. While FID is a reasonable choice given the research focus, relying solely on FID without additional measures is problematic. Running multiple trials and reporting the standard deviation of FID scores could improve the robustness of the evaluation.

**Other Comments Or Suggestions:**

.

**Other Strengths And Weaknesses:**

The paper is well-written.

**Questions For Authors:**

In Figure 7, (a) shows 1-RF achieving an FID of 3.01, yet (b) achieves 2.58 and (d) achieves 2.61.
Does this suggest that the baseline result of 3.01 was an outlier due to variance in training or sampling?
Would running more trials and reporting standard deviations clarify whether 3.01 was a statistical anomaly?

**Relation To Broader Scientific Literature:**

The ideas could be extended in conditional diffusion models (with varying classifier-free guidance settings) and latent diffusion models.

**Theoretical Claims:**

The theoretical statements are mostly reasonable, but the paper mixes different types of diffusion models without clearly distinguishing them. Table 1 properly differentiates DDIM, EDM, and FM, but
In Statement 1 and 2, the focus is on FM, while Section 5 is heavily EDM-oriented. Since Table 2 shows that DDIM, EDM, and FM behave differently under noise-unconditioning, a general claim that "DMs work without noise conditioning" oversimplifies the issue. Instead, the results suggest that the effects of noise-unconditioning depend on the underlying diffusion formulation, requiring a more detailed explanation that is currently missing.

---

> ### Author Rebuttal · Authors · 2025-04-01
>
> Thank for the constructive feedback and supportive comments. Here, we address the concerns regarding classifier-free guidance (CFG), experimental random variance, model-wise analysis, and explanation on performance of 1-RF.
>
> **1. Latent Diffusion (DiT) & Classifier-free Guidance**
>
> Reviewer 7y3v commented, `One issue is that diffusion sampling often involves classifier-free guidance, which introduces instability through extrapolation in `$x_t$`. Noise-unconditioning may exacerbate these instabilities`.
>
> This comment on the risk of unstability of generation with classifier-free guidance is very valuable. Also, we believe that incorporating larger-scale experiments such as Latent Diffusion (DiT) will definitely make our work more solid.
>
> To address the concern, we conducted experiments DiT + FM (i.e. SiT) on the ImageNet 256x256 dataset with CFG. Here, all experiments use the same configuration as the original paper, using Euler sampler with 250 steps. The results demonstrate that our findings extend successfully to larger scales:
>
> | Model: DiT-B/2 + FM |  |  |
> | --- | --- | --- |
> | CFG Scale | FID w/ t | FID w/o t |
> | 2.0 | *9.36* | 10.66 |
> | 2.5 | _**8.03**_ | 8.15 |
> | 2.7 | 8.24 | _**7.96**_ |
> | 3.0 | 8.88 | _8.15_ |
> | 3.5 | 10.29 | _9.09_ |
> | 4.0 | 11.81 | _10.28_ |
>
> Notably, removing noise conditioning improves performance at optimal CFG scales. Also, for many different CFG scales, removing noise conditioning improves performance. We can see that in the setting of FM on large-scale dataset, the behavior of removing noise conditioning remains consistent with experiments in our paper.
>
> These results show that it’s feasible to extend our conclusions to large-scale diffusion models, and our observations are robust with respect to classifier-free guidance.
>
> **2. Statistical Robustness of the Results**
>
> **Reviewer 7y3v:** `In Figure 7, (a) shows 1-RF achieving an FID of 3.01, yet (b) achieves 2.58 and (d) achieves 2.61. Does this suggest that the baseline result of 3.01 was an outlier due to variance in training or sampling? `
>
> To address this concern, we report the variance of FID on 5 trials with different random seeds (the top row), as well as using three different training checkpoints (the bottom row). Results are as follows. These numbers correspond to the last row in Figure 7(a).
>
> | Model: FM (1-RF) | (a) | (b) | (c) | (d) |
> | --- | --- | --- | --- | --- |
> | different sample seeds | $3.01\pm 0.02$ | $2.58\pm 0.02$ | $2.65\pm 0.02$ | $2.61\pm 0.01$ |
> | different checkpoints | $2.98\pm 0.03$ | $2.62\pm 0.03$ | $2.66\pm 0.01$ | $2.61\pm 0.02$ |
>
> Note that here a, b, c, d are different settings.
>
> These results confirm that the performance enhancement is not due to a statistical anomaly. The generation quality measured by FID-50k is robust enough for the evaluation.
>
> **3. Over simplification?**
>
> **Reviewer 7y3v:** `In Statement 1 and 2, the focus is on FM , while Section 5 is heavily EDM-oriented… oversimplifies the issue`.
>
> We clarify that our analysis is valid for general cases. While we used FM formulation as our primary example, our analysis extends to all listed diffusion models. We chose FM for its notational simplicity and clarity, avoiding unnecessary complexity while maintaining theoretical rigor.
>
> **4. Explanations on 1-RF: Curvature of the Path**
>
> We greatly appreciate your insightful perspective on how 1-RF inherently aligns with the diffusion ODE trajectory.
>
> Our theory mainly focuses on the bounding the error of removing noise conditioning, demonstrating that most models can work with the removal of noise-conditioning. We have to modestly admit that our theory is incapable of explaining why 1-RF becomes better when the noise-conditioning is removed.
>
> Nevertheless, we believe that using the curvature of trajectory to build connection between the noise-unconditional model’s performance to specific diffusion formulations is a very promising future direction, and we would like to explore in more depth on this topic in follow-up works.

---

> > ### Comment · Reviewer_7y3v · 2025-04-04
> >
> > Thank you for the solid responses to points 1 and 2. I appreciate the clear answers and additional experiments.
> >
> > Regarding points 3 and 4, I still believe that the differences between FM and other diffusion models in how they respond to the removal of noise conditioning deserve deeper investigation. Exploring these differences could lead to a more complete understanding and a stronger contribution overall.
> >
> > I see point 3 as a key weakness of the current version of the paper. While the authors seem to agree that point 4 might offer a potential direction to address this limitation, it wasn't directly addressed in the rebuttal. For these reasons, I would like to maintain my score.

---

> > > ### Author Response · Authors · 2025-04-07
> > >
> > > Thank you for your thoughtful comments and for recognizing the value of our responses to points (1) and (2). We are glad that the additional experiments and variance analysis were helpful and provided further support for our findings, making our results more solid and comprehensive.
> > >
> > > Regarding points (3) and (4), we appreciate the insightful suggestions. Our theoretical analysis is intended to be broadly applicable across diffusion formulations, and FM was used as a representative example for clarity. We acknowledge that model-specific behaviors deserve deeper investigation, and such exploration will represent a promising direction for future work. Also, while our theory does not directly explain why 1-RF improves without noise conditioning, we agree that understanding such behavior — possibly through geometric perspectives like trajectory curvature — is an exciting avenue to pursue going forward.
> > >
> > > We sincerely thank you again for the constructive feedback. We have made concrete efforts to address the concerns through new experiments, statistical validation, and clarification of theoretical scope. We also engaged thoughtfully with the broader conceptual points raised, and hope our responses reflect the care with which we approached this work.

---

### Official Review · Reviewer_gvd2 · 2025-03-13

**Overall Recommendation:** 3

**Summary:**

The paper tries to debunk a common belief among diffusion model practitioners if a time-condition of the model is necessary for a diffusion model. The authors take both the theoretical and experimental approach to address this issue. The paper mainly focus on the theoretical reasoning rather than practice, which is only demonstrated by CIFAR-10 generation problem, although various samplers are chosen for demonstration.

**Claims And Evidence:**

I believe that the paper’s main questioning on the necessity of the time-conditioning is valid and might open a new area of research that will benefit generative model communities. I am content with the theoretical justification of the paper and the choice of various sampling mechanisms. However, I am highly suspicious that CIFAR-10 generation task and commercial large-scale latent diffusion models do not share the same level of difficulty, and extending the “success” in CIFAR-10 domain to the large scale LDM should be done in extreme care. I still believe that the paper’s main question is well addressed. However, unless we have an actual experimental result for large scale commercial diffusion models, the practical value of this claim is not strong enough. I do not oppose to rejecting this so far, and I want to hear about other’s opinions regarding this including the authors’.

**Essential References Not Discussed:**

It would be nice to include DPM-Solver, DPM-Solver++-type samplers as well, since these are still well-used samplers in practice, and also these samplers use multiple tabbed filtering. The performance for these type of samplers might be closely related to DDIM's failure as well.

**Experimental Designs Or Analyses:**

It is a redundant statement, but I am consent with the discussion of various different samplers. However, I am not very well consent about the domain of generation.

**Methods And Evaluation Criteria:**

As I have mentioned in the previous section, the claim for diffusion model should be done in extreme care and should be equipped with practical large scale latent diffusion models of any kind (e.g., Stable Diffusion, or other image/audio/text diffusion models). Unless any one of these experiments are conducted I cannot give higher score than 3. I do not think we need every single large scale experiments but I only think that CIFAR-10 is not enough, regarding the small size of domain, the simplicity of the manifold, and the extreme sparsity of the dataset compared to the large scale data such as LAION-5B.

**Other Comments Or Suggestions:**

- The figures seems not to address the main claim of the paper well enough. For example, in Figure 1, I have looked into multiple times to realize that the core idea is that diffusion models may just work well with t removed. This type of presentation issues exist in every figures in the manuscript and I recommend to update those with better clarity.

**Other Strengths And Weaknesses:**

Overall, the paper gives an important question on the necessity of the time condition dependency of the model. This claim reveals that the architecture of current diffusion models may only be suboptimal.

I am also happy that the paper is equipped with rich theoretical justification and experimental justification with various types of samplers.

Regarding weakness claims, please refer to other sections of this review.

**Questions For Authors:**

- Is there any practical problem to adopt this method in large scale diffusion models?

**Relation To Broader Scientific Literature:**

The problem tackles the necessity of noise-level condition for multi-step denoising frameworks. If the paper’s claim is valid in general denoising problems, this can be extended to deep learning-based solutions for ill-posed problem in general.

**Theoretical Claims:**

I have found no error in the theoretical claims. However, I would like to humbly admit that I may have missed some errors, though.

---

> ### Author Rebuttal · Authors · 2025-04-01
>
> We sincerely appreciate the thoughtful feedback and recognition of our theoretical contributions. Below, we address the concerns regarding experimental scope and practical applicability to large-scale diffusion models.
>
> **1. Large-scale experiment**
>
> To address reviewer gvd2’s concerns on experimental scale, we conducted additional large-scale experiments with DiT + FM (i.e. SiT) [1, 2] on ImageNet 256x256 dataset. All experiments use the same configuration as the original paper, using Euler sampler with 250 steps. The results demonstrate that our findings extend successfully to larger scales:
>
> | Model: DiT-B/2 + FM |  |  |
> | --- | --- | --- |
> | CFG Scale | FID w/ t | FID w/o t |
> | 2.0 | *9.36* | 10.66 |
> | 2.5 | _**8.03**_ | 8.15 |
> | 2.7 | 8.24 | _**7.96**_ |
> | 3.0 | 8.88 | _8.15_ |
> | 3.5 | 10.29 | _9.09_ |
> | 4.0 | 11.81 | _10.28_ |
>
> Notably, removing noise conditioning improves performance at optimal CFG scales. Also, for many different CFG scales, removing noise conditioning improves performance. We can see that in the setting of FM on large-scale dataset, the behavior of removing noise conditioning remains consistent with experiments in our paper.
>
> **2. Performance on Other Types of Samplers**
>
> **Reviewer gvd2:** `“It would be nice to include DPM-Solver, DPM-Solver++-type samplers as well, since these are still well-used samplers in practice, and also these samplers use multiple tabbed filtering. The performance for these type of samplers might be closely related to DDIM's failure as well.”`
>
> This is a very interesting extension to our current work, generalizing our theory and experiment to a wider range of commonly used sampling methods.
>
> Regarding the DPM solver equation:
>
> $$
> x_{t_i}=\frac{\alpha({t_i})}{\alpha(t_{i-1})}x_{t_{i-1}}-\sigma({t_i})(e^{h_i}-1)\epsilon_{\theta}(x_{t_{i-1}},t_{i-1})
> $$
>
> (Eq. 3.7 in [3]), which, in our notation (Eq. 4 in our paper), becomes
>
> $$
> \kappa_i = \frac{\alpha_{i+1}}{\alpha_{i}},\quad \eta_i = -\sigma_{i+1}(e^{h_{i+1}}-1),\quad \zeta_i=0
> $$
>
> This $\kappa_i$ for DPM solver behaves very similar with DDIM (which has $\kappa_i=\sqrt{\frac{\alpha_{i+1}}{\alpha_i}}$), which leads to large error bound value (the term $\prod \kappa_i$ dominates the error bound, and  $\prod \kappa_i$ approaches infinity due to division by 0). Thus, we can **expect** a bad performance similar with DDIM in the noise-unconditional scenario by our theory.
>
> To further confirm this intuition, we perform numerical calculations of bound values as well as FID evaluations for both DPM-Solver and DPM-Solver++ (which has a similar formulation as DPM-Solver but uses $x$-prediction). The bound value of both of them are on the order of $10^6$ (which is of the same order with DDIM), and using DPM solver fails to generate good images (FID $>50$).
>
> These results shows that our theoretical error bound (in Section 4.4) can qualitatively predict the performance accurately, demonstrating that our theory generalizes on even these carefully designed samplers.
>
> **3. Other Suggestions**
>
> We appreciate the feedback regarding figure clarity and will incorporate the suggestions to enhance the presentation of our results.
>
> **References**
>
> 1. Peebles, William, and Saining Xie. "Scalable diffusion models with transformers." *Proceedings of the IEEE/CVF international conference on computer vision*. 2023.
> 2. Ma, Nanye, et al. "Sit: Exploring flow and diffusion-based generative models with scalable interpolant transformers." *European Conference on Computer Vision*. Cham: Springer Nature Switzerland, 2024.
> 3. Lu, Cheng, et al. "Dpm-solver: A fast ode solver for diffusion probabilistic model sampling in around 10 steps." *Advances in Neural Information Processing Systems* 35 (2022): 5775-5787.
> 4. Lu, Cheng, et al. "Dpm-solver++: Fast solver for guided sampling of diffusion probabilistic models." *arXiv preprint arXiv:2211.01095* (2022).

---

> > ### Comment · Reviewer_gvd2 · 2025-04-08
> >
> > Thank you for the comment.
> >
> > I have read the other reviews and the authors responses on them, too. I believe this paper is a well-established work that questions the necessity of time-conditioning in diffusion models. It would be better to extend the discussion in Section 4.4 on why DDIM and DPM-Solvers fail with this method and why the other samplers gain advantage even if the noise level condition is removed. I will maintain my score as weak acceptance.

---

### Official Review · Reviewer_iMKL · 2025-03-14

**Overall Recommendation:** 5

**Summary:**

This paper analyzes noise conditional diffusion models (DMs) and develops theory supporting the viability of noise unconditional DMs. Empirical evidence supports the author's theoretical claims and demonstrates that noise unconditional DMs are capable of performance similar to noise conditional DMs. This challenges pre-existing notions that noise conditioning is fundamentally necessary for DMs to function.

**Claims And Evidence:**

Yes, all claims in the paper are well supported both by theoretical proofs and empirical studies.

**Essential References Not Discussed:**

The finding in Section 4.1 (i.e., that the expectation over multiple realizations of z is an effective target) is very reminiscent of Noise2Noise [1]. Including a reference to [1] would provide additional context around using the expectation over multiple noisy realizations as a target.

**References**

1. Lehtinen, Jaakko, et al. "Noise2Noise: Learning image restoration without clean data." arXiv preprint arXiv:1803.04189 (2018).

**Experimental Designs Or Analyses:**

Yes, the empirical studies appear valid to me.

**Methods And Evaluation Criteria:**

Yes.

**Other Comments Or Suggestions:**

N/A

**Other Strengths And Weaknesses:**

**Strengths**
1. Novelty. I find the author's idea to use multiple realizations of z as an effective target to train an unconditional DM to be novel.
2. The theory and proofs explore the viability of noise unconditional DMs, as well as their error bounds
3. Impact. The theory is well supported by empirical evidence and unconditional DMs (somewhat surprisingly) perform similarly to noise conditional DMs. This is an important finding for the community because it challenges pre-existing notions about DMs.

**Questions For Authors:**

N/A

**Relation To Broader Scientific Literature:**

This paper challenges the existing notion that noise conditioning is fundamental to the high performance of DMs.

**Theoretical Claims:**

I have reviewed the proofs / theoretical claims in the main paper and they appear sound to me.

---

> ### Author Rebuttal · Authors · 2025-04-01
>
> Thanks for the thoughtful review and positive feedback regarding our work’s novelty and impact!
>
> We agree that incorporating the suggested reference *Noise2Noise: Learning image restoration without clean data* will provide valuable context for the derivation in section 4.1. We will revise the manuscript to include this reference.

---

### Decision · Program_Chairs · 2025-05-01

**Decision:**

Accept (poster)

**Comment:**

This paper argues that noise conditioning is not as vital as previously thought to obtain good performance with diffusion models. The reviewers all praised the theoretical contributions and empirical results on CIFAR-10, but criticized the lack of experiments on higher-dimensional data. The authors provided such experiments in their rebuttal, assuing the reviewer's concerns. I thus recommend acceptance.